

# Capturing the signature of heavy rainfall events using the 2-d-/4-d
# water vapour information derived from GNSS measurement
# in Hong Kong
**Qingzhi Zhao [1,*], Yibin Yao [2] and Wanqiang Yao [1]**
[1] College of Geomatics, Xi'an University of Science and Technology, Xi'an 710054, China;
zhaoqingzhia@163.com; sxywq@163.com
[2] School of Geodesy and Geomatics, Wuhan University, Wuhan 430079, China; ybyao@whu.edu.cn
* Correspondence: zhaoqingzhia@163.com; Tel.: +86-182-9185-5186
**Abstract:** Apart from the well-known applications like positioning, navigation and timing (PNT),
Global Navigation Satellite System (GNSS) has manifested its ability in many other areas that are
vital to society largely. With the dense setting of the regional continuously operating reference
station (CORS) networks, monitoring the variations in atmospheric water vapour using a GNSS
technique has become the focus in the field of GNSS meteorology. Most previous studies mainly
concentrate on the analysis of relationship between the two-dimensional (2-d) Precipitable Water
Vapour (PWV) and rainfall while the four-dimensional (4-d) variations of atmospheric water
vapour derived from the GNSS tomographic technique during rainfall events are rarely discussed.
This becomes the focus of this work, which investigates the emerging field of GNSS technology
for monitoring changes in atmospheric water vapour during rainfall, especially in the vertical
direction. This paper includes an analysis of both 2-d, and 4-d, precipitable water vapour profiles.
A period with heavy rainfall events in this study was selected to capture the signature of
atmospheric water vapour variation using the ground-based GNSS tomographic technique. GNSS
observations from the CORS network of Hong Kong were used. Analysed results of the 2-d
PWV/4-d water vapour profiles change during the arrival, occurrence, and depression of heavy
rainfall show that: (i) the PWV time series shows an increasing trend before the arrival of heavy
rainfall and decreases to its average value after the depression of rainfall; (ii) rainfall leads to an
anomalous variation in relative humidity and temperature while their trends are totally opposite
and show daily periodicity for periods without rain (this is highly correlated with the changes in
solar radiation); (iii) atmospheric water vapour presents unstable conditions with intense vertical



convective motion and hydrometeors are formed before the arrival of rainfall while returning to
relatively stable conditions during heavy rainfall. This study indicates the potential for using
GNSS-derived 2-d PWV and 4-d profiles to monitor spatio-temporal variations in atmospheric
water vapour during rainfall, which provides a better understanding of the mechanism of
convection and rainfall induced by the extreme weather events.
**Keywords:** GNSS; PWV; water vapor profiles; extreme weather events

**1 Introduction**
Precipitable water vapour (PWV), which refers to the total content of integrated water vapour
density along the zenith direction, is a significant component reflecting the short-term atmospheric
water vapour variations used in severe weather detection as well as in long-term climate studies
(Bai, 2004; Liu et al., 2013); however, it is difficult to obtain a satisfactory spatio-temporal
resolution of atmospheric water vapour due to the limitation of both the number of traditional
sounding stations and the observation times (Brenot et al., 2013; Zhang et al., 2015). For the past
20 years, the ability to estimate water vapour contents with an accuracy of 1 to 2 mm has been
proved using the Global Navigation Satellite System (GNSS), which generally formed a new field
of study in GNSS Meteorology (Bevis et al., 1992). Therefore, the variation of atmospheric water
vapour with high accuracy, as well as the high spatio-temporal resolution can be obtained using
the hyper-dense GNSS networks (with receivers only a few kilometres apart).
PWV, at high spatio-temporal resolution is an indicator for monitoring the water vapour responses
to severe weather events (Zhang et al., 2015; Yao et al., 2017; Zhao et al., 2018a, 2018b). It has
been used for operational meteorology in some areas such as Japan (JMA, 2013), the UK (Bennitt
and Jupp, 2012), France (Guerova et al., 2016), and Italy (Barindelli et al., 2018). In those areas,
the zenith total delays (ZTD) or PWV estimated from ground-based GNSS measurements are
generally assimilated into numerical weather prediction (NWP) models (De Haan 2013; Saito et
al., 2017). In addition, ZTD or PWV is also used for the early warning and forecasting of severe
precipitation, which has been investigated in areas of Greater Lisbon in Portugal as well as
Zhejiang Province in China (Benevides et al., 2015; Yao et al., 2018; Zhao et al., 2018a, 2018b).
These applications have verified the ability of GNSS as used in meteorology, but those cases are
mainly focussed on two-dimensional (2-d) PWV which cannot reflect the specific vertical





variations in atmospheric water vapour.
Although GNSS tropospheric tomography has been proposed (Flores et al., 2000), and can be used
to obtain four-dimensional (4-d) water vapour variations, the development of this technique has
mainly focussed on improvement of theoretical and model aspects while its application is rarely
discussed. For example, the reliability of GNSS tomography was validated using radiosonde data
by Seko et al. (2000) and Troller et al. (2002, 2006). The joint reconstruction of atmospheric water
vapour was also investigated by combing multi-GNSS observations as well as multi-source data
derived from the Constellation Observing System for Meteorology, Ionosphere, and Climate
(COSMIC), Interferometric Synthetic Aperture Radar (InSAR), radiosonde flights, *etc*. (Bender
and Raabe, 2007; Bender et al., 2011; Wang et al., 2014; Alshawaf, 2013; Heublein et al., 2015;
Benevides et al., 2015; Zhao et al., 2018c). For the improvement of tomographic models and
resolution thereof, Perler et al. (2011) proposed a new parameterised tomographic method, which
is capable of obtaining better tomographic results. Some methods concerned with the resolution of
tomographic models, as well as the division of tomography areas, have been proposed such as the
extended sequential successive filtering method, iterative reconstruction algorithm, *etc*. (Braun et
al, 2003, 2004; Wang et al., 2014; Zhao et al., 2017a, 2018d; Chen and Liu, 2014). In addition,
maximal use of GNSS signals penetrating from the side faces of tomography areas has obtained a
significant improvement and is realised by introducing the water vapour scale factor (Yao and
Zhao, 2016; Yao et al., 2016; Zhao et al., 2017b).
Currently, GNSS tomography technique is maturing in terms of theoretical and model aspects
through almost 20 years of development, but its application in GNSS meteorology remains to be
further investigated, therefore, we focus on capturing the signature of heavy rainfall events using
the 2-d/4-d water vapour information derived from GNSS measurements in Hong Kong. The 2-d
PWV time series is first analysed for correlation with heavy rainfall. Thereafter, the signatures of
4-d water vapour variations derived from GNSS tomography are investigated during heavy rainfall
events while the tomographic modelling is resolved using the optimal weighting determination
method.

**2 Fundamentals of GNSS meteorology**

**2.1 Retrieval of GNSS PWV**





Satellite signals are delayed and bent when crossing the atmosphere, which adds ionosphere and
troposphere delay: the former delay can be eliminated based on ionosphere free (IF) linear
combination during the processing of GNSS measurement due to the dispersive nature of
ionosphere delay (Dach and Walser, 2013). The latter delay can be divided into two parts:
hydrostatic delay and wet delay. The first part of the tropospheric delay in a vertical direction, also
called zenith hydrostatic delay (ZHD), can be precisely calculated by the Saastamoinen model
(Saastamoinen, 1972) with the observed surface pressure while the second part can be estimated in
the zenith direction using GNSS data. The second part is also called zenith wet delay (ZWD),
from which the PWV can be calculated, thus forming a new concept: GNSS meteorology, as first
proposed by Bevis et al. (1992). The calculation used in obtaining PWV is expressed as follows:
the zenith total delay is first estimated by processing the GNSS measurements using the GNSS
processing software such as Bernese, GAMIT, *etc*. The ZWD is then obtained by extracting the
ZHD from ZTD and thus the PWV can be calculated based on the following equations
(Saastamoinen, 1972; Askne and Nordius, 1987; Bevis et al., 1992):

$$
\begin{aligned}
&\mathrm{PWV} = \Pi \cdot \mathrm{ZWD} \\
&\Pi = 10^6 \Big/ \Big( (k_2^{'} + k_3 / \mathrm{Tm}) \cdot R_v \cdot \rho_w \Big) \\
&\mathrm{ZWD} = \mathrm{ZTD\text{-}ZHD} \\
&\mathrm{ZHD} = \frac{0.002277 \times \mathrm{P}}{1 - 0.00266 \times \cos(2\varphi) - 0.00028 \times \mathrm{H}}
\end{aligned}
\tag{1}
$$


Where $\Pi$ refers to the conversion factor, where $k_2^{'}$, $k_3$, and $R_v$ are constants with values of
22.1 K/mb, $3.739 \times 10^5$ K$^2$/mb and 461.495 J/kg/K, respectively, $T_m$ represents the weighted mean
temperature, which is related to surface parameters such as temperature and pressure. Therefore,
$T_m$ is usually calculated based on the empirical model using the data from radiosonde or numerical
weather model due to the observed layered meteorological parameters with are rarely obtained
(Bevis et al., 1994; Yao et al., 2012). In the fourth formula in Eq. (1), $\mathrm{P}$, $\mathrm{H}$, and $\varphi$ represent
the surface pressure (hPa), geodetic height (km), and station latitude (rad), respectively. In our
study, the value of $T_m$ is calculated based on the established regional $T_m$ model using the
radiosonde data and observed temperature (Section 3.2).

**2.2 Establishment of tomographic model**





Generally, the slant wet delay (SWD) or slant water vapour (SWV) is considered as the input
information for GNSS troposphere tomography (Flores et al., 2000; Hirahara, 2000; Skone and
Hoyle, 2005; Rohm and Bosy, 2009; Chen and Liu., 2014) and the following equation gives an
expression used to obtain SWV (Flores et al., 2000):
$$\text{SWV}_{azi,ele} = m_w(ele) \cdot \text{PWV} + m_w(ele) \cdot cot(ele) \cdot (G_{NS}^{w} \cdot cos(azi) + G_{WE}^{w} \cdot sin(azi)) \qquad (2)$$
Where $m_w$ presents the wet mapping function. $ele$ and $azi$ refer to the elevation angle and
azimuth angle, respectively. $G_{NS}^{w}$ and $G_{WE}^{w}$ are the gradient parameters in the south-north and
west-east directions, respectively.
If a sufficient number of SWVs derived from some stations in a regional CORS network can be
obtained, the GNSS tomographic technique can be used to reconstruct the three-dimensional (3-d)
distribution of atmospheric water vapour field. Therefore, a four-dimensional (4-d) water vapour
information is a time series of such a 3-d tomographic result, which can reflect the regional
atmospheric water vapour variations in both the spatial and temporal domains. As described by
Flores et al. (2000), the linear observation equation between SWV and water vapour density can
be expressed as follows:
$$\text{SWV} = \sum (d_{ijk} \cdot x_{ijk}) \qquad (3)$$
Where $i, j, k$ represent the location of the area of interest in the longitudinal, latitudinal, and
vertical directions, respectively, $d_{ijk}$ and $x_{ijk}$ refer to the distance travelled by satellite signals
and the water vapour density remains to be estimated, respectively in the discretized voxels
$(i, j, k)$. Therefore, the matrix form of the tomographic observation equation can be described as
follows:
$$\mathbf{y} = \mathbf{A} \cdot \mathbf{x} \qquad (4)$$
Where $\mathbf{y}$ represents the column vector of SWV derived from GNSS measurements. $\mathbf{A}$ and $\mathbf{x}$
are the coefficient matrix of distance penetrated by satellite rays and the column vector of water
vapour density, respectively.
Due to the large sparse matrix of observation equation, some constraints are required to overcome
the influence caused by the ill-posed problem in the inversion of the tomographic normal equation
(Flores et al., 2000; Bi et al., 2006; Bender et al., 2011; Rohm and Bosy, 2011 Chen and Liu,



2014). In our study, both horizontal and vertical constraints are considered. The water vapour
density in a certain voxel is regarded as the weighted mean value of its horizontal neighbouring
voxels (Rius et al., 1997) and the negative exponential function is introduced to describe the
relationship between the nearby voxels in the vertical direction while the coefficients of functional
model are established using radiosonde data (Yao and Zhao, 2016). Consequently, the
tomographic modelling can be expressed after imposing the constraints as:

$$
\begin{pmatrix} \mathbf{y} \\ \mathbf{0} \\ \mathbf{0} \end{pmatrix} = \begin{pmatrix} \mathbf{A} \\ \mathbf{H} \\ \mathbf{V} \end{pmatrix} \cdot \mathbf{x}
\tag{5}
$$


Where $\mathbf{H}$ and $\mathbf{V}$ are the coefficient matrices of horizontal and vertical equations, respectively.
To obtain a reasonable tomographic result from the above equation, an optimal tropospheric
solution method is used, which can adaptively tune the weightings of different types of equations
(Zhao et al., 2018d).

**3 Data description and establishment of a regional $T_m$ model**
**3.1 Data description**
To validate the ability of GNSS technique in capturing the signature of atmospheric water vapour
variation during heavy rainfall events, two periods of GNSS observations (19 to 27, July 2015 and
1 to 8, August 2015) from 13 GNSS stations in the CORS network of Hong Kong are selected in
the experiment. Those two periods are selected because they correspond to a heavy rainfall event
and a no-rainfall event, respectively according to hourly rainfall data from 45 rain gauges evenly
distributed across this area (Figure 1). There is a radiosonde station located in this area where the
radiosonde balloon is launched twice daily at UTC 00:00 and 12:00, respectively. The 20-years of
radiosonde data from 1998 to 2017 are used to establish the regional $T_m$ model in this study. In
addition, the surface temperature and relative humidity are also selected to analyse their changes
during those two periods. To explain the variations of surface temperature and relative humidity,
the solar radiation data are also used in this study, which is derived from the CRU-NCEP Ver. 7
dataset. This dataset is a combination product of the CRU TS3.2 climate dataset and the NCEP
reanalysis data. The temporal-spatial resolution of the solar radiation dataset are four times daily
(UTC 00:00, 06:00, 12:00 and 18:00) and 0.5°×0.5°, respectively.



GNSS observations are processed using Precise Point Positioning (PPP) data processing software
and the accuracy of the estimated ZTD parameters has been proved with the values of 7.2 mm and
8.1 mm when compared to the GAMIT (v10.5) and Bernese (v5.2) software, respectively (Zhao et
al., 2018a). The sampling rate of the estimated ZTD is 30 s and the data processing strategy has
been presented previously (Zhao et al., 2018d). In addition, the gradient parameters in south-north
and east-west directions are also estimated at intervals of 2 h. The corresponding meteorological
parameters, such as the surface pressure and temperature, are also obtained at the selected GNSS
stations. Therefore, the precise ZHD can be calculated by the empirical model using the observed
surface pressure. The conversion factor, as described in Eq. (1), is also obtained, in which $T_m$ is
calculated based on the established $T_m$ model which will be introduced in the following section.
Finally, the PWV time series, as well as the SWVs for the 13 selected GNSS stations, can be
obtained. Five of the 45 rain gauges (R21, TMS, PEN, SSP, and KSC) are selected to analyse the
variations in atmospheric water vapour during different weather conditions (Figure 1).

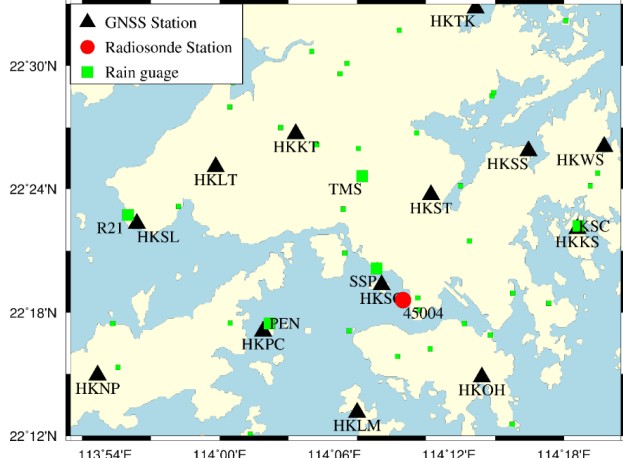


Figure 1. Geographic distribution of selected GNSS and radiosonde stations as well as the rain
gauges used in the experiment

**3.2 Establishment of the regional $T_m$ model**
Due to the layered parameters such as water vapour pressure, temperature, *etc.* generally cannot be
obtained for the location of GNSS stations, the $T_m$ values of those stations are calculated based on
the empirical model in this experiment. It has been proved that $T_m$ is highly correlated with the



variations of temperature, pressure, and the seasons (Bevis et al., 1992; Yao et al., 2012; Yao et al.,
2014, 2015; Liu et al., 2018). Therefore, a regional $T_m$ model which includes as parameters:
temperature, surface pressure, and seasonal variation, is established and expressed as follows:

$$T_m = T_{m0} + a*T_s + b*P_s + c*\cos(2\pi\frac{doy}{365.25}) + d*\sin(2\pi\frac{doy}{365.25})$$
$$+ e*\cos(4\pi\frac{doy}{365.25}) + f*\sin(4\pi\frac{doy}{365.25}) \tag{6}$$

Where $T_{m0}$, $T_s$, and $P_s$ represent the initial value $T_m$, surface temperature, and surface pressure,
respectively; $doy$ refers to the day of year; $a$ and $b$ are coefficients of $T_s$ and $P_s$,
respectively, while $c$ to $f$ refer to the coefficients of the seasonal correction function. In our
study, the coefficients in Eq. (6) were obtained by the least squares regression method using
20-year radiosonde data series for 45004 while the values of $a$ to $f$ are 129.1225, 0.5370,
-0.0023, 0.358, 0.813, -0.178, and 0.255, respectively.
The performance of the established $T_m$ model is analysed and compared with the empirical
formula proposed by Bevis et al. (1994). Statistical result of 20-years of radiosonde data reveals
that the standard deviation and bias for the established $T_m$ model and the empirical formula
proposed by Bevis et al. (1994) are 2.04/0.0009 K and 3.41/2.53 K, respectively, which indicates
that the established regional $T_m$ model is superior to the empirical formula. The further to analyse
the impact of $T_m$ model error on the calculated PWV, a comparison experiment is carried out for
radiosonde station 45004 with a variation in $T_m$ of 1 K, 3 K, 5 K, 7 K, and 9 K, respectively and
compared with the actual PWV values. Figure 2 shows the impact of $T_m$ error on PWV for
radiosonde station 45004 with a change in $T_m$ of 1 K, 5 K, and 9 K, respectively. It can be clearly
seen from Figure 2 that the impact of $T_m$ model error on PWV is negligible. Statistical analysis
shows that the PWV errors induced by the change in $T_m$ of 1 K, 3 K, 5 K, 7 K, and 9 K are 0.15
mm, 0.45 mm, 0.75 mm, 1.04 mm, and 1.34 mm, respectively under the condition of PWV > 0
mm, while the values are 0.18 mm, 0.54 mm, 0.91 mm, 1.27 mm, and 1.63 mm, respectively when
PWV > 40 mm. Therefore, the PWV errors caused by the established $T_m$ model in this study are
less than 0.4 mm and 0.5 mm when PWV > 0 mm and PWV > 40 mm, respectively. Such result is
deemed acceptable for the analysis of PWV variations with rainfall events (Akilan et al., 2015).




Figure 2. Impact of Tm to PWV for radiosonde station (45004) with a change in Tm by 1 K, 5 K
and 9 K, respectively over the period of 1998 to 2017

## 4 Signature of 2-d/4-d variations in atmospheric water vapour during rainfall

According to the recordings of 45 rain gauges derived from the Hong Kong Observatory, it is
continuous rains in Hong Kong for the period of 19 to 27, July 2015 with the largest rainfall more
than 300 mm. The weather conditions are cloudy and sunny without rainfall happened for the
period of 1 to 8, August 2015. Therefore, those two periods are selected in this paper to investigate
the variation characteristics of atmospheric water vapor.

### 4.1 Cases of 2-d PWV time series change

To capture the signature of PWV time series change in different weather conditions, the
comparison between the 5-minute GNSS-derived PWV and hourly rainfall are performed for the
periods of 19 to 27, July 2015 and 1 to 8, August 2015, respectively. Four GNSS stations (HKKS,
HKSC, HKPC, and HKSL) and the surrounding rainfall gauges (HSC, SSP, PEN, and R21) are
selected for this experiment.
Figure 3 shows the variations of 5-minute PWV time series data with hourly rainfall as well as the
cumulative rainfall at those four stations for the period of 19 to 27, July 2015 with its frequent
rainfall events. It can be seen, from Figure 3, that the PWV time series show an increasing trend
before the arrival of rainfall and reaches a relatively large value during rainfall, PWV then returns
to its average value after rainfall. Additionally, the PWV time series data present a downward



trend at four stations during this period. The cumulative rainfall first increased at about UTC 11:00,
20 July, 2015 with different levels reached and the event terminated at UTC 12:00, 23 July, 2015.
The largest cumulative rainfall reached about 250 mm while the minimum recorded rainfall was
about 100 mm across the four selected gauge stations. The PWV time series is also analysed at
those four stations for the period from 1 to 8, August, 2015 in which no rainfall was recorded
(Figure 4). Figure 4 shows the 5-minute PWV time series changes from which it can be found that
PWV does not show any continuous increasing trend when there is no rainfall, but the range of
PWV variation is relatively large (from about 35 mm to greater than 55 mm). Comparing the
PWV time series in Figures 3 and 4, it also can be observed that the PWV values during rainfall
are much larger than that of no rainfall time.
In addition, 5-minute surface temperature and relative humidity data are also analysed during
those two periods. The first and second columns of Figure 5 show the changes in temperature and
relative humidity for the period 19 to 27, July, 2015. It also can be seen that the temperature and
relative humidity do not show any trend during heavy rainfall but show a tendency to run counter
to one another on 19, 26, and 27, July. one explanation is that heavy rainfall breaks the trend in
temperature and relative humidity for the period from 20 to 25, July, 2015. The third column of
Figure 5 shows the changes in solar radiation for this period, from which it can be observed that
the solar radiation undergoes a day periodic change. To verify this explanation, the variations of
temperature and relative humidity, as well as those in solar radiation, are also presented at those
four stations for period without rainfall (Figure 6): temperature and solar radiation show a similar
trend while relative humidity presents the opposite trend. Additionally, it can be observed from the
first and third columns of Figures 6 that the maximum values of solar radiation and temperature
occurred at UTC 4:00 (local time 12:00) while the minimum value of relative humidity also
occurred at that time. The phenomenon found in Figure 6 further confirmed the explanation
presented above. In addition, the values of solar radiation are more fluctuated at the four stations
during rainfall when compared to that without rain (Figures 4 and 6): a possible reason for this is
that the part of solar radiation is decreased by cloud cover during heavy rain.





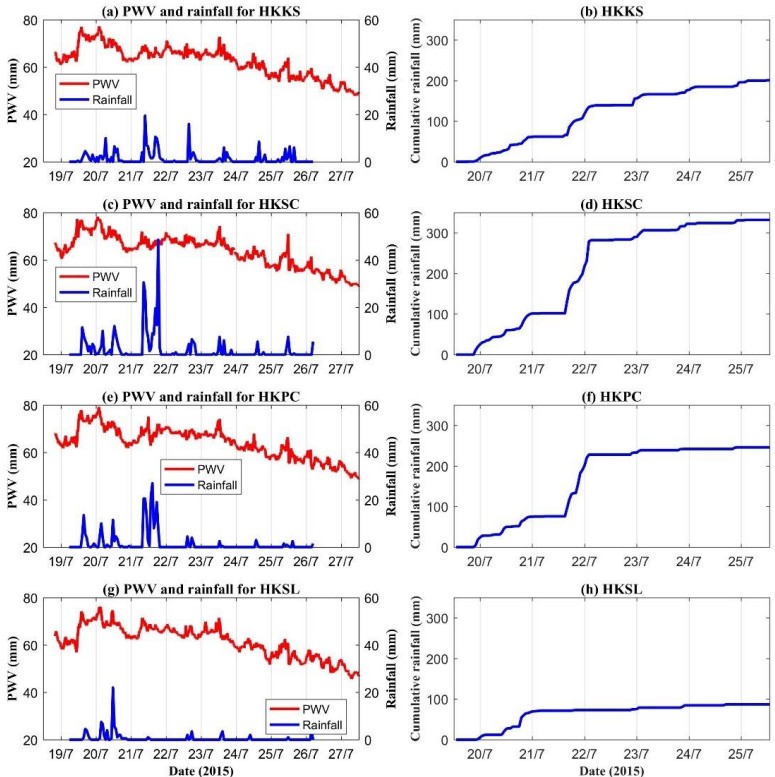


Figure 3. Variations of 5-minutely PWV time series with hourly rainfall and the cumulative
rainfall for HKKS, HKSC, HKPC and HKSL stations over the period of 19 to 27, July 2015, the
first column represents the variations of PWV and rainfall and the second column refers to the

cumulative rainfall

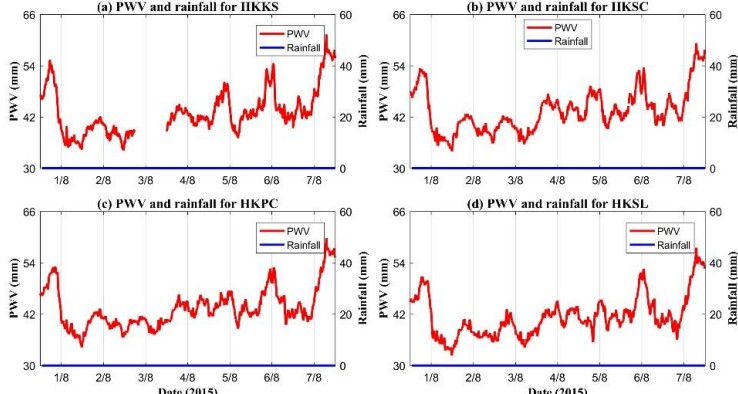


Figure 4. Variations of 5-minutely PWV time series with hourly rainfall at (a) HKKS, (b) HKSC,



(c) HKPC and (d) HKSL stations over the period of 1 to 7, August 2015


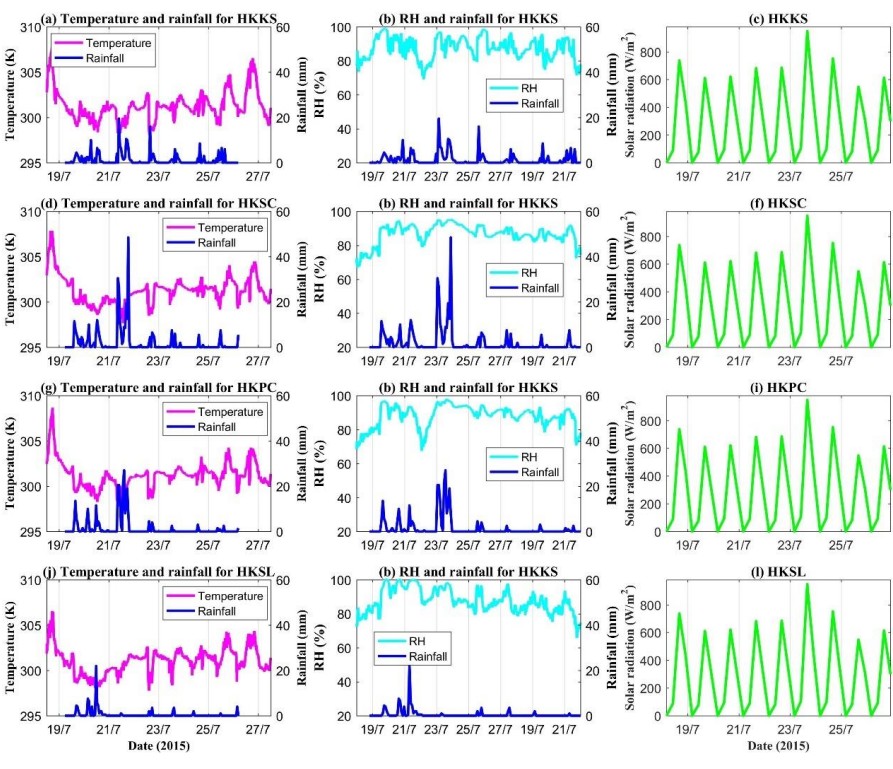



Figure 5. Changes of temperature, relative humidity with rainfall as well as the solar radiation at
HKKS, HKSC, HKPC and HKSL stations over the period of 19 to 27, July 2015, the first column
represents the variations of temperature and rainfall, the second column refers to the variations of

RH and rainfall and the third column refers to the solar radiation




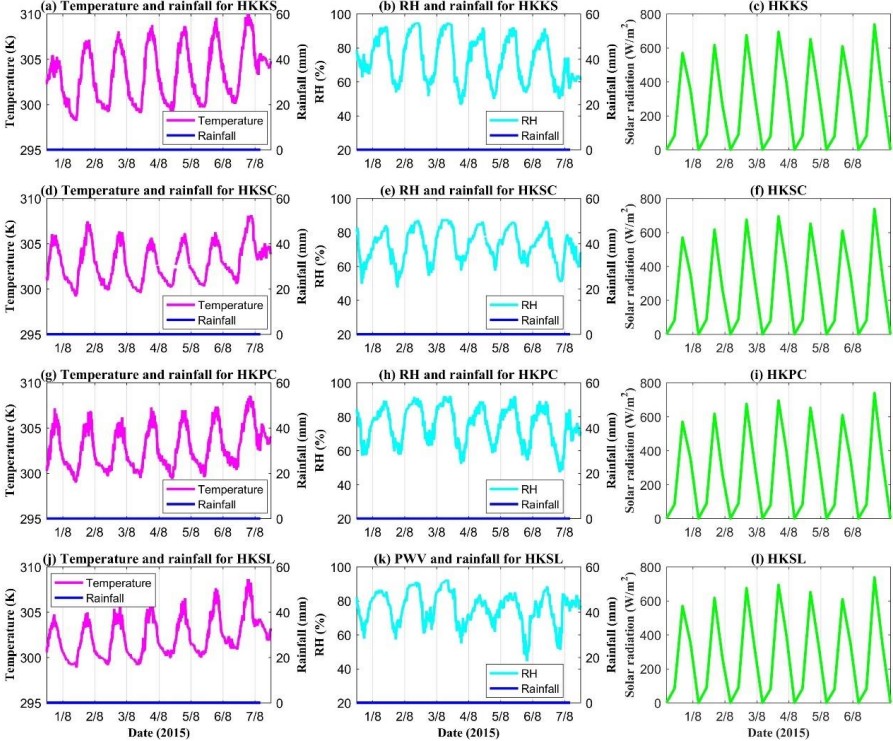

Figure 6. Changes of temperature, relative humidity with rainfall as well as the solar radiation at HKKS, HKSC, HKPC and HKSL stations over the period of 1 to 7, August 2015, the first column represents the variations of temperature and rainfall, the second column refers to the variations of RH and rainfall and the third column refers to the solar radiation

**4.2 Cases of water vapour profile variation during heavy rainfall**

The variations in 4-d atmospheric water vapour are also analysed during heavy rainfall. In this section, the tomographic technique is introduced and the research area is discretised. There are 7 and 8 grids in longitudinal and latitudinal directions, respectively and 29 layers in vertical direction. Therefore, there are total $7 \times 8 \times 29$ voxels. The horizontal steps are 0.05° and 0.06° in longitudinal and latitudinal directions, respectively while the inhomogeneous vertical step is selected based on the water vapour distribution at different altitudes (Yao and Zhao, 2017) with resolutions of 0.2 km × 10, 0.3 km × 8, 0.4 km × 6, 0.6 km × 4, and 0.8 km × 1, respectively. A





comparison of water vapour density profiles derived from tomographic result and radiosonde data
at the location of radiosonde station 45004 is first presented (Figure 7) to validate the performance
of the GNSS tomographic technique. It can be seen from the Figure 7 that the profiles derived
from tomographic results were consistent with the observed radiosonde data at most altitudes,
which manifests the ability of the GNSS tomographic technique to reflect variations in water
vapour content during rainfall. The detailed information about the accuracy of tomographic result
has been presented in Zhao et al. (2018d). In addition, it also can be observed that more
sophisticated water vapour variations detected vertically (with 29 layers) can be provided by the
GNSS tomographic technique than by radiosonde data.
Two heavy rainfall periods are selected in this experiment: the first at UTC 18 to 22, 21 July 2015
and three rain gauges are used to analyse the variations in water vapour profiles. The hourly
rainfall for those three rain gauges is presented in Table 1 while the water vapour profile variations
over time for SPP, PEN, and TKL are shown in Figures 8 to 10. From those three figures it can be
observed that atmospheric water vapour profile undergoes vertical movement about 1-2 hours
before the arrival of heavy rain, which is reflected by the fluctuating water vapour density at
different altitudes. For the SPP rain gauge, it can be seen that the water vapour content in the
lower atmosphere, from an altitude of about 1.8-2.5 km to 3.5 km while the water vapour content
decreases from 4-5 km to 3.5 km. It can also be observed from PEN and TKL rain gauges that an
upward and downward movement happened in the atmospheric water vapour profile in the lower,
and upper atmosphere, respectively: this results in a large increase in atmosphere water vapour at
altitudes of about 2.3 km and 1.6 km, respectively (especially at PEN). The upward and downward
motions of atmospheric water vapor in the lower and upper atmosphere are expected to the
occurrence of the strong convective weather. In addition, it was found that the variations of water
vapour profiles in vertical direction at station TKL are weaker than that from stations PEN and
SPP. A possible explanation is that the rainfall was 30.5 mm and 20.5 mm for PEN and SPP at
UTC 20, 21 July 2015 while the value is only 1 mm at station TKL at UTC 21, 21 July 2015
(Table 1). The above phenomenon indicates that the significant vertical motion of water vapour
profile was possibly induced by heavy rainfall. The variations in water vapour profiles during
rainfall reveal that the significant vertical motion of water vapour occurred before the onset of
rainfall while the water vapour profiles were relatively stable during rainfall events.





In addition, the time series of water vapour density profiles, at a temporal resolution of 1 minute,
for the three rain gauges are also presented in Figure 11. From which it can be seen that the
vertical water vapour density profile undergoes a significant vertical motion about 1-2 hours
before the arrival of rain (black dotted rectangles, Figure 11) while the profiles are relatively
stable during rain. By comparing the Figures 8-10, it also can be found that the vertical variations
of water vapour density profiles at SPP and PEN stations 1 hour before rainfall are more active
than that at TKL station: this can be explained by considering that the continued heavy rainfall
happened at SPP and PEN stations while the TKL had little rainfall (Table 1), therefore, the
continuing water vapour transportation in the vertical direction existed in the lower atmosphere at
stations SPP and PEN.

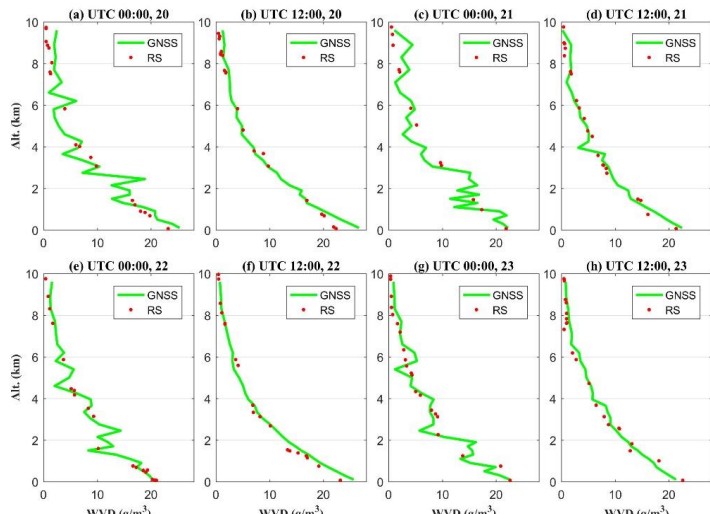

Figure 7. Distribution of water vapor density (WVD) profiles at UTC 00:00 and 12:00,
respectively derived from the GNSS tomographic result (green curve) and the radiosonde data of
the observed height (red hot) for the location of radiosonde station (45004) over the period of 20
to 23, July 2015

Table 1 Hourly Rainfall information of the selected four rain gauges over period of UTC 19 to 23,
21 July 2015 (Unit: mm)

| Station  Date | SPP | PEN | TKL |
|---|---|---|---|





| | | | |
|---|---|---|---|
| 19, 21 July | 0 | 0 | 0 |
| 20, 21 July | 30.5 | 20.5 | 0 |
| 21, 21 July | 26.5 | 20.5 | 1.0 |
| 22, 21 July | 10.5 | 13.5 | 1.5 |
| 23, 21 July | 7.5 | 2.5 | 1.5 |


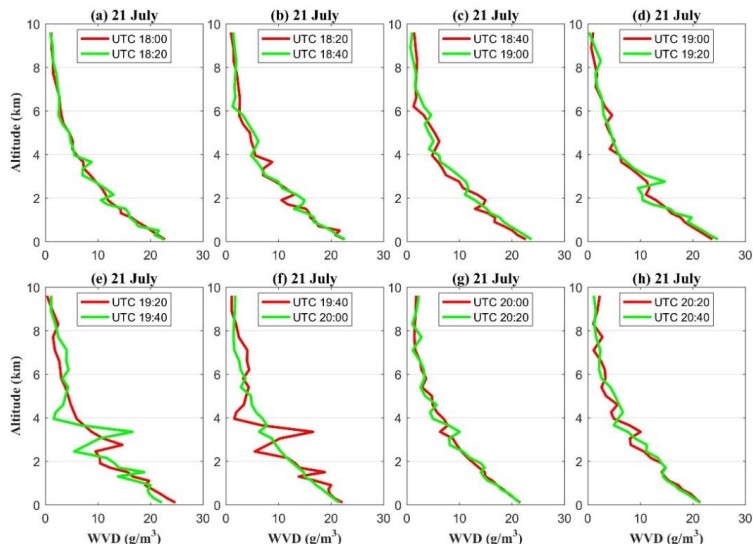


Figure 8. Distribution of water vapor density Profiles (WVD) derived from GNSS tomographic

result with the temporal resolution of 20 minutes for the location of SPP rain gauge over the

period of UTC 18:00 to 20:40, 21 July 2015






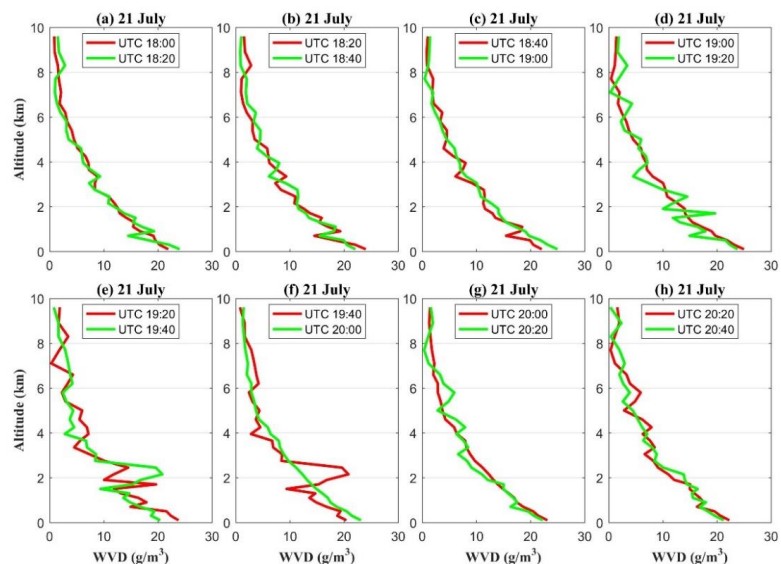

Figure 9. Distribution of water vapor density Profiles (WVD) derived from GNSS tomographic

result with the temporal resolution of 20 minutes for the location of PEN rain gauge over the

period of UTC 18:00 to 20:40, 21 July 2015


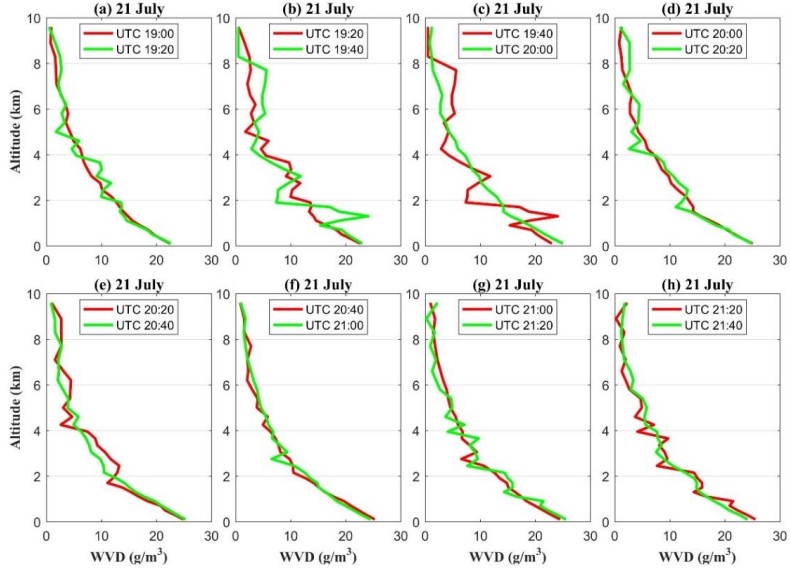

Figure 10. Distribution of water vapor density Profiles (WVD) derived from GNSS tomographic

result with the temporal resolution of 20 minutes for the location of TKL rain gauge over the

period of UTC 19:00 to 21:40, 21 July 2015





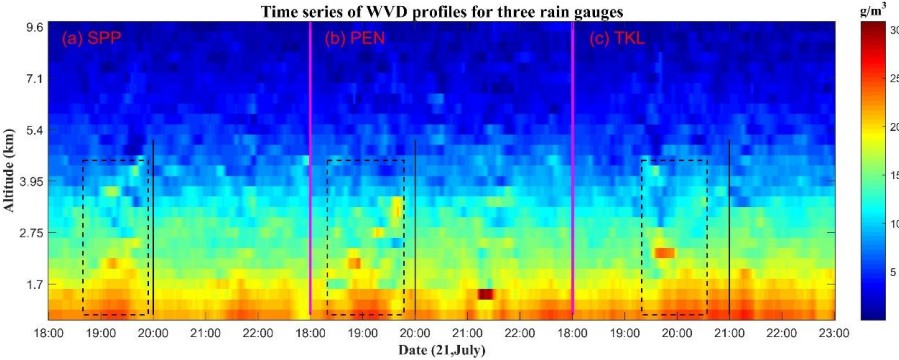


Figure 11. Time series of water vapor density (WVD) profiles derived from GNSS tomographic

result with the temporal resolution of 1 minute for the locations of three rain gauges over the

period of UTC 18:00 to 23:00, 21 July 2015, where the rainfall happened at UTC 20:00 21 July

for (a) SPP and (b) PEN rain gauges while the rainfall occurred at UTC 21:00 21 July for (c) TKL

rain gauge. The WVD profiles with drastic vertical motion are marked by the black dotted

rectangles for three rain gauges while the locations of black solid lines are the starting time of

rainfall


To verify the phenomenon observed above, another period (UTC 0 to 4, 23 July 2015) at PEN and
TMS rain gauges is selected while the hourly rainfall information is presented in Table 2. Figures
12 and 13 both reflect that the change in water vapour profiles at PEN and TMS stations are
similar to that of above conditions. The water vapour content above PEN and TMS is increased at
altitudes of 2.5 km and 3.2 km, respectively, some 1-2 hours before onset of rainfall and returns to
its average value at the moment that the rainfall is about to begin. One possible explanation for
this is that: before onset of rainfall, the atmospheric water vapour was conditionally unstable with
intense vertical movement as proved by Brenot et al., (2006). The ascending motion of water
vapour in the lower atmosphere and the descending motion of water vapour in the upper
atmosphere significantly increases the water vapour content at a certain height where
hydrometeors are formed. The hydrometeors consist of liquid water and icy hydrometeors,
formation of which is random in time and space. Due to the delays to satellite signals induced by
liquid water and icy species generally being much smaller than the water vapour species-induced
delays, these are unavailable in the case of GNSS observations, therefore, GNSS tomography



cannot reflect the distribution of hydrometeors and the tomographic profiles show a
returning-to-the-mean trend after the formation of hydrometeors. These newly-generated
hydrometeors particles form raindrops with a continual accretion thereof. When the atmosphere is
unable to support the weight of the formed raindrop, the drop falls as rain. The formation of
hydrometeors particles and raindrops require some time, hence the intense vertical movement of
atmospheric water vapour before onset of rainfall. The time taken to generate hydrometeors and
raindrops provides the possibility of now-casting rainfall based on the GNSS technique.

Table 2. Hourly Rainfall for the selected four rain gauges over the period UTC 1 to 5, 23 July

2015 (Unit: mm)

| Station Date | PEN | TMS |
|---|---|---|
| 0, 23 July | 0 | 0 |
| 1, 23 July | 0 | 0 |
| 2, 23 July | 4.5 | 16.5 |
| 3, 23 July | 0.5 | 4.5 |
| 4, 23 July | 0 | 0.5 |


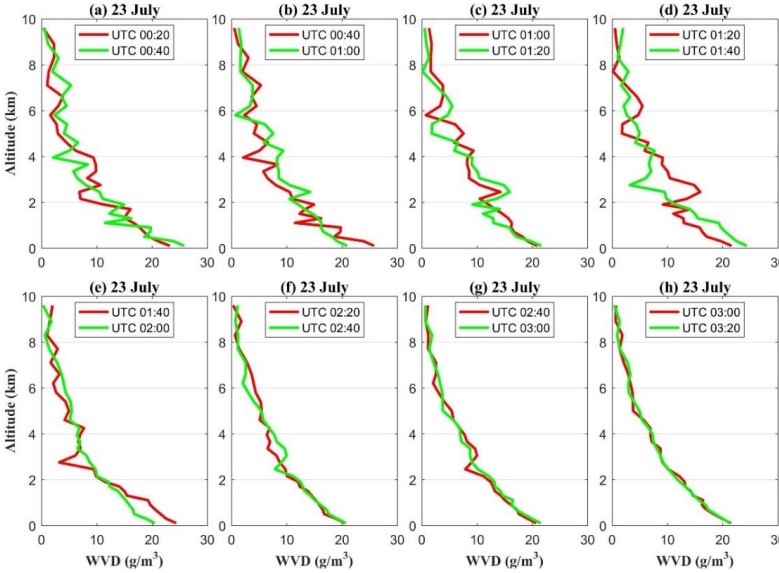


Figure 12. Distribution of water vapor density (WVD) profiles from GNSS tomographic result
with the temporal resolution of 20 minutes for the location of PEN rain gauge over the period of



UTC 00:20 to 03:20, 23 July 2015


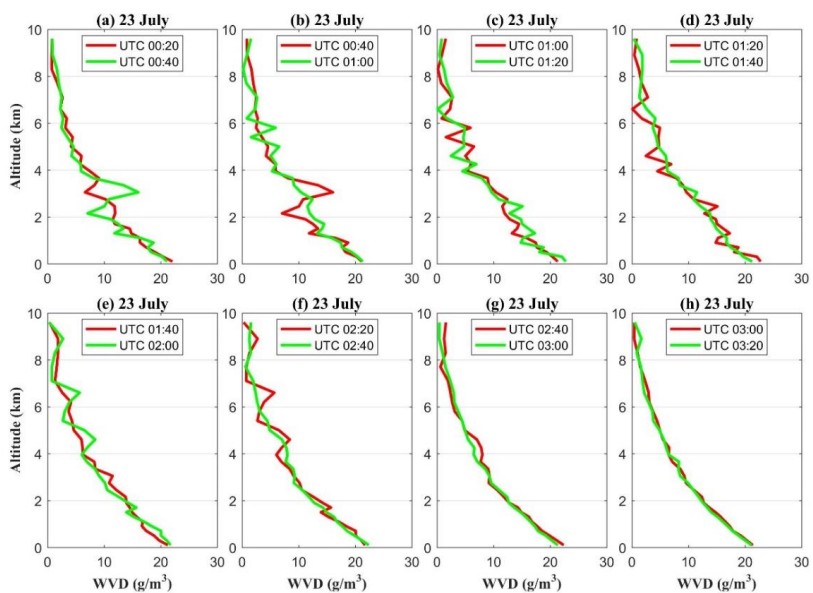


Figure 13. Distribution of water vapor density (WVD) profiles from GNSS tomographic result

with the temporal resolution of 20 minutes for the location of TMS rain gauge over the period of

UTC 00:20 to 03:20, 23 July 2015


The 4-d distribution of atmospheric water vapour for the period UTC 18:00 to 20:20, 21 July 2015

is presented with a spatio-temporal resolution of 20 minutes and 20 layers to an altitude of 5 km,

respectively (Figure 14). According to the hourly rainfall recordings at 45 rain gauges in this area,

most parts of the experimental area suffered heavy rainfall at UTC 20:00, 21 July 2015 that lasted

for several hours. It can be found, from Figure 14, that the significant vertical motion of water

vapour observed over the period from UTC 18:00 to 19:40 returns to its relatively stable condition

at UTC 20:00 but with a lower water vapour content in most layers. The main reason for this may

be water vapour transfer to the liquid water particles and icy hydrometeors, which have little

impact on the delay of satellite signals and cannot be observed by the GNSS technique. For the

period of heavy rainfall that occurred after UTC 20:00, the atmospheric water vapour profiles

were relatively stable with slight vertical variation in water vapour content. In addition, it can be

concluded that the place at which hydrometeors were generated in the lower atmosphere is



possibly where rainfall occurred. Therefore, where heavy rainfall occurred is possibly predictable
before the onset of rainfall according to the 4-d atmospheric water vapour variations at different
altitudes derived from GNSS tomography. It also can be found that there is the horizontal motion
of atmospheric water vapor as well in different layers, especially at the bottom layers. This is
because the happening of rainfall requires the enough water vapor supplement, the horizontal
motion of water vapor at the bottom layers implies the continuous water vapor transportation.

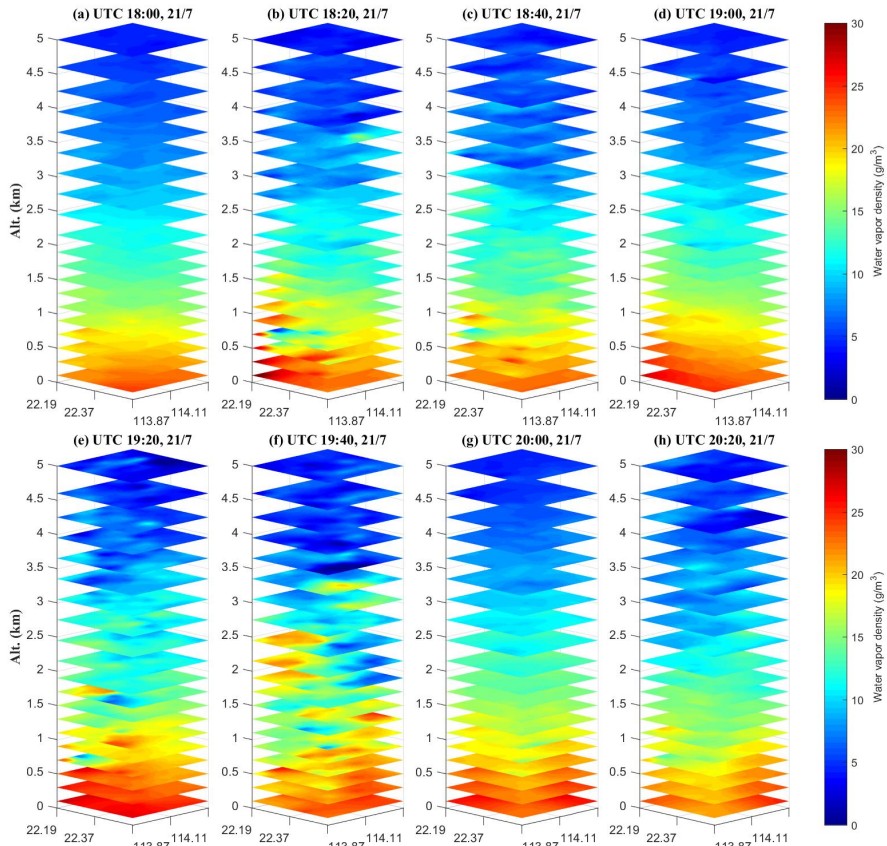


Figure 14. Three-dimensional distribution of atmospheric water vapor density derived from GNSS

tomographic result with the temporal resolution of 20 minutes over the period of UTC 18:00 to

20:20, 21 July 2015 with 20 layers from the ground to 5km



**5 Conclusion**



GNSS sensing water vapour is an effective, practical technique, which able to the reflect 2-d and
4-d atmospheric water vapour variations during the formation and lifecycle of heavy rainfall. 2-d
PWV time series data derived from GNSS observations are first compared with hourly rainfall
measurements, which reveals the continuous increasing trend in PWV before the onset of rainfall
and returns to its average value after rainfall. In addition, it is also found that the variations of
surface temperature and relative humidity have day-periodicity and are mainly caused by the
variations in solar radiation during no rain periods, but their changes are disturbed by rainfall
during rainfall periods.
A 4-d water vapour reconstruction technique is performed using GNSS data to analyse the vertical
water vapour movement during rainfall period. It is found that significant vertical motion occurred
about 1-2 hours before the arrival of rainfall and this was reflected by the ascending and
descending motions of water vapour in the lower and upper atmosphere, respectively.
Hydrometeors are then formed at a certain altitude where sufficient water vapour was concentrated.
The formation of hydrometeors and raindrops requires some time, which makes it possible for the
forecasting of now-casting rainfall. At the moment of onset of rainfall, the water vapour profiles
return to their average values at different altitudes and show a relative stable condition but with a
decreasing trend in the water vapour content in the lower atmosphere. In addition, the place where
the rainfall is most possible happened may be forecasted by locating out the location of the point
of decreasing water vapour content in the lower atmosphere. These results revealed that rainfall
had a direct relationship with atmospheric water vapour content as well as the vertical variations
of water vapour density profiles, which further manifested the significant potential of the GNSS
technique for monitoring and forecasting during the lifecycle of rainfall event.


**Acknowledgement:** The authors would like to thank IGAR for providing access to the web-based
IGAR data. The Lands Department of HKSAR is also acknowledge for providing GNSS and
meteorological data from the Hong Kong Satellite Positioning Reference Station Network (SatRef)
and the corresponding rainfall data. This research was supported by the Key projects of National
Natural Science Foundation (4P179511).



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
