# Peer review of "Capturing the signature of heavy rainfall events using the 2-d-/4-d"

_Annales Geophysicae, 2018_

## Referee Comment (RC1) · Anonymous Referee #1 · 13 Sep 2018

The work described in the manuscript is generally worth publishing however the manuscript itself requires major revisions to be accepted - both on the side of content and the way how are the results presented.

General comments

1. Based on your findings you provide very strong statements about a superiority of your GNSS results however evidences for it are often either very poor or completely missing. Except radiosonde profiles (which seem to have somehow limited vertical

resolution) you do not use any reference product which would support the results provided by your GNSS tomography. You also do not describe at all the meteorological situation itself – what type of precipitation occurred (convective, stratiform), how was it developed, etc. Generally, you directly link the increase of water vapour or its vertical movement to a formation of hydrometeors and consequent rainfall. Although this can be potentially correct, there can be situations where an increase of water vapour or its vertical movement will not lead to any precipitation – just because the rainfall life cycle is not related only to water vapour as it is a much more complex process. Have you checked this? I strongly recommend you to discuss your results with somebody who has strong knowledge in meteorology since I miss this knowledge. It would allow you to much better justify your results. And please see my major comment 6 and 7 for more information regarding this general comment.

2. The overall quality of some of your figures is rather poor and it is really not easy to follow and interpret them. Therefore, I recommend you to do edit some of your figures in the below given major comment 5.

3. Although the quality of written English is not bad, some issues occur occasionally. I recommend you to let a native speaker proofread your manuscript before a next submission.

Major comments

1. I have some comments on how you compute your values of SWV (equation 2, P5L121): - Please use the term horizontal tropospheric gradients instead of just gradients and introduce their meaning

- It would be worthy to at least mention that horizontal tropospheric gradients represent a gradient of ZTD, not just of ZWD. Are you aware of this? Although during the periods you describe in your study the prevailing gradient was probably the gradient of water vapour, it would be possibly worthy to subtract the hydrostatic part from the total gradient

- Probably you are aware, that you should use gradient mapping function for mapping the gradients to the elevation angle of the observation, not the wet mapping function (mw). Although your formula corresponds to Bar-Sever gradient mapping function, you should explicitly state it, because there are other gradient mapping functions based on different formulas. In the manuscript you don't mention anywhere which mapping function you did use for hydrostatic and wet components.

- Did you use the conversion factor kappa (which was used to convert ZWD to PWV) also to scale the original values of gradients to "PWV gradients"? I ask because according to your eq. 2 you did not do that and

- Have you considered using post-fit residuals for SWV computation? If not you should at least mention their existence since during severe weather events they can contain important information about tropospheric water vapour distribution which cannot be captured by ZTD or gradients (see i.e. Kačmařík et al., 2017)

2. P6L169: Are you sure that solar radiation data provided by a global model with a 0.5x0.5 ° horizontal resolution and 6h time interval of outputs is a reasonable source of solar radiation data for the level of local meteorological events you work with? And maybe even more importantly: it is absolutely clear that solar radiation is dependent on day/night change and an occurrence of clouds and that it influences the temperature and relative humidity. Why do you include it in your study, what exactly you want to show using it?

3. P7L173: I miss important information about your GNSS data processing:

- Which PPP software did you use? Which mode (I guess post-processing), precise products, mapping function, cut-off elevation angle, etc. did you use?

- Are you sure that you estimated ZTD every 30 s (what was then the observation time interval)? Usually while a deterministic modelling of tropospheric parameters is used, the ZTD is estimated in a 5-minute or longer interval (and on P9L232 you also mention

that you used PWV in 5-minute interval). Does the software used for your processing is based on a deterministic or a stochastic modelling of troposphere? Since you state that horizontal tropospheric gradients were estimated in 2-hour interval, I guess that it used deterministic approach. Why have you chosen this type of setting to estimate ZTD every 30 s, but gradients only every 2 h?

- What is meant with the presented "accuracy" of estimated ZTD parameters? Is it standard deviation or root-mean-square error or any other statistical parameter? Although you don't provide the information on what exactly these numbers represent, I don't consider 7 or 8 mm as high quality ZTD estimates. For example, the official IGS ZTD product is stated to have an overall accuracy of 4 mm (Byram et al., 2011). I didn't check the referenced paper of Zhao et al. (2018d), however I would suggest you to provide more information on what these values represent and what kind of solutions using Gamit or Bernese were used for these comparisons.

4. P8L208: Could you please explain what presented values of bias/standard deviation mean? Is it a, a comparison between Tm from your regional and standard empirical model or b, a comparison between Tm computed from radiosonde profiles and Tm from your regional and standard empirical model? I guess the b, is right however it is not fully clear. Anyway I would be careful with your statement that shown results indicate that "the established regional Tm model is superior to the empirical formula". Because if you used radiosonde profiles to correctly establish your regional model, then it MUST be very close to the actual radiosonde profiles. So you only proofed that the established model should provide good results in your area (supposing the radiosonde data are considered to be error-free).

5. Comments regarding selected figures:

- Figure 2: since the differences in PWV estimated using different Tm are very small, it is practically impossible to see anything in the figure. Therefore, I propose not to include the figure at all and only optionally provide some statistical information about

the variation of PWV based on various Tm values.

- Figure 3, 4, 5, 6: It is really hard to a, see something in detail in these figures, b, compare results in figures 3, 4 with results in figures 5 and 6. In different words, it is really complicated to confirm your written description and interpretation of these figures. Therefore, I strongly recommend you to 1, increase the size of these figures, 2, provide detailed looks on interesting periods (i.e. these with the highest rainfall), 3, try to put all the shown parameters into one figure per station (I mean combine information from figures 3 and 5 and from figures 4 and 6, for example show temperature and Rh together with PWV and rainfall in just one figure – the individual curves can be shifted using a constant offset to increase the readability). I also recommend you to use the same scale in axis y in all figures to make their mutual comparison fair (i.e. in figure 3 you use for PWV interval from 20 to 80 mm, but in figure 4 an interval from 30 to 66 mm).

- Figures 8, 9, 10: I recommend you to provide all these figures as ONE figure with ONE caption. I also recommend you to somehow mark time of a beginning of the precipitation in these figures to increase their readability.

6. Section 4.1 – I miss a reasonable discussion of your results together with their short summary. I can see only a description of your figures together with some generally valid information (like that the solar radiation is connected with day/night cycle or cloud coverage or that the IWV time series can exhibit an increase of values before the beginning of rainfall). What is your interpretation of whole shown time series? Is there any clear relation between PWV and any other meteorological parameter? Are your results in agreement with other researches who studied this topic? Can PWV time series help us to predict rainfall? You should deal with questions like these. Please see also my minor comments related to individual sentences in section 4.1.

7. Section 4.2

- P14L306: I would really correct your statement in sentence "In addition, it also can be

observed that more sophisticated water vapour variations detected vertically (with 29 layers) can be provided by the GNSS tomographic technique than by radiosonde data." Firstly, looking on figure 7 I consider your radiosonde profiles not to have a full vertical resolution which is standardly accessible by this instrument (standardly you should get one measurement per approximately 30 m). Could you please explain why your radiosonde profiles are so coarse? Secondly, even if your GNSS tomography could reach a better vertical resolution than a radiosonde (what is not possible with just 29 vertical layers), you could not call your tomography profiles to be "more sophisticated" – because you are not able to proof that variation in your profiles is related to real meteorological conditions and not only to errors of tomographic reconstruction.

- P18L382: I don't agree with you that "delays to the satellite signals induced by liquid water and icy species ... are unavailable in the case of GNSS observations". The GNSS signals ARE influenced by them (see Solheim et al., 1999 and Kačmařík et al., 2017) and therefore the estimated ZTD/SWD/SIWV contain these effects. However, it is not possible to separate them from the estimated parameters. This also means that your GNSS tomography profiles should be influenced by these hydrometeors.

- I don't see any trustworthy quality evaluation either for your SWV values or for the vertical profiles from GNSS tomography reconstruction. How you can justify that described changes in presented vertical profiles of WV are related to real weather conditions and not only to mismodelling deficiency of your GNSS tomography solution? You should at least discuss this topic.

- Please see also my minor comments related to individual sentences in section 4.2.

Minor comments

1. P2L42: you write about using GNSS PWV for severe weather and climate studies, however you provide only two references. Since there are many studies related to GNSS meteorology and (a) severe weather monitoring (i.e. Japanese team around Y. Shoji); (b) climate (i.e. Gradinarsky et al., 2002, Vey et al., 2010, Ning a Elgered 2012,

Bock et al., 2014), I recommend you to cite at least the most important ones and cite them separately for (a) and (b).

2. P2L44: what do you mean with "traditional sounding stations" – radiosondes? It should be explicitly given. Are you aware of WV observations from remote sensing satellites or of other instruments as WVR or Raman Lidar?

3. P2L50: I would rather write that GNSS PWV can be used to provide information about water vapour distribution, which is related to form of precipitation and not only to severe weather events. In this regard – how do you define a severe weather event in your perspective?

4. P2L52: I would rather write that GNSS PWV is operationally used for their assimilation into numerical weather prediction models (NWM) than just for operational meteorology. Italy doesn't use GNSS PWV operationally, work presented in Barindelli et al. (2018) was just a case study to promote an operational deployment.

5. P2L56: I would rather say that ZTD or PWV can be used for early warnings than that it is used. I recommend you here to cite the work of Brenot et al. (2013) at least.

6. P3L65: there are much more (recent) studies that used radiosonde profiles for GNSS tomography validations, i.e. Shangguan et al. (2013)

7. P3L68: COSMIC is not an instrument for water vapour sensing, it is whole program designed for various purposes – in this regard I would rather write GNSS Radio occultation technique to be consistent.

8. P3L75: I would not say that iterative reconstruction techniques deal with resolution of tomography models or division of tomography areas.

9. P4L91: please correct your statement that ionosphere causes signal delay – since ionosphere causes a delay for code measurements, but an advance for phase measurements (see i.e. Hofmann-Wellenhof et al. 2008). Also you cannot fully eliminate ionosphere with just the IF linear combination – this eliminates only the first order effect,

but not effects of higher order.

10. P4L95: I would firstly introduce the hydrostatic and non-hydrostatic part of the delay and then call the non-hydrostatic the wet delay. Also please use zenith direction all the time (instead of a vertical direction)

11. P4L101: replace the word measurements with a more appropriate signals

12. P4L102: provide references for given processing software

13. P4L103: you should rather mention that in the GNSS data processing the ZHD is usually taken from an a priori model and later precisely computed from i.e. real meteorological observations to subtract ZWD from ZTD. Because strictly speaking in the GNSS data processing you do not estimate a ZWD, but a correction to a priori modelled ZHD.

14. P4L107: use rather base SI units for all the given coefficients. It would be also worthy to mention that different sets of refractivity constants exist (see Rueger, 2002)

15. P5L117: I recommend replacing the term SWV with SIWV (Slant integrated water vapour)

16. P5L142: Rohm (2013) developed a GNSS tomography solution using no constraints, so please correct your sentence according to that

17. P6L169: could you provide a reference for given CRU-NCEP solar radiation data sets you used in your study?

18. P7L191: with the "layered parameters" you mean that information about vertical profile of some meteorological parameters as water vapour pressure or air temperature is not available? I recommend you rewriting this sentence to make better understandable what you mean.

19. P8L209: I recommend you to see the paper of Ning et al. (2016) providing a rigorous evaluation of uncertainty in GNSS IWV estimation including impact of Tm

uncertainty. I would also like to put you into perspective with the number of overall achievable accuracy of IWV estimates from GNSS which is meant to be around 0.4 – 0.6 mm (see Guerova et al., 2016) – it would be possible good to mention it in the paper.

20. P9L227: 300 mm is a value for one day and one station? Or for whole studied period from July 19 till July 25? It is not clear from your sentence.

21. P9L240: with the sentence "Additionally, the PWV time series data present a downward trend at four stations during this period" you mean the whole processed period from June 19 till July 25 or anything else? If yes, why do you state it? It is from whatever reason important for your study?

22. P10L244: please add information at which stations these values of cumulative rainfall occurred.

23. P10L249: what do you want to say with your sentence that "the PWV values during rainfall are much larger than that of no rainfall time" You just want to state the fact, that this situation occurred in your selected time periods or you want to state it as a general fact which is valid every time? Because I would not agree with the second option.

24. P14L314: The sentence starting "For the SPP rain gauge" is not understandable for me.

25. P14L325: I would consider rewriting the sentence starting "The above phenomenon indicates". The heavy rainfall could be induced by vertical motion of water vapour, but the opposite (variation of water vapour induced by heavy rainfall) is not so logical.

26. Table 1 and Table 2: could you provide a more detailed look on the precipitation? I mean provide the rainfall information not only in 1-hour interval, but for example in 5-minutes or 10-minutes time interval. You don't need to provide it in a table, in can be in a figure (optionally in the figures 8, 9, 10 themselves or in figure 11). It would provide

the reader a much better idea how was the rain spread out in time.

References

Bock, O., Willis, P., WaDng, J., Mears, C. A high-quality, homogenized, global, long-term (1993–2008) DORIS precipitable water data set for climate monitoring and model verification, Journal of Geophysical Research, 119, 7209–7230, doi:10.1002/2013JD021124, 2014.

Brenot, H., Neméghaire, J., Delobbe, L., Clerbaux, N., Meutter, P., Deckmyn, A., Delcloo, A., Frappez, L., Van Roozendael, M. Preliminary signs of the initiation of deep convection by GNSS, Atmospheric Chemistry and Physics, 13, 5425–5449, doi:10.5194/acp-13-5425-2013, 2013.

Byram, S., Hackmann, C., Tracey, J. Computation of a highprecision GPS-based troposphere product by the USNO, Proceedings of the 24th International Technical Meeting of The Satellite Division of the Institute of Navigation (ION GNSS 2011), Portland, USA, September 19-23, 2011.

Gradinarsky, L. P., Johansson, J., Bouma, H. R., Scherneck, H. G., Elgered, G. Climate monitoring using GPS, Physics and Chemistry of the Earth, 27, 335–340, doi:10.1016/S1474-7065(02)00009-8, 2002.

Guerova, G., Jones, J., Douša, J., Dick, G., de Haan, S., Pottiaux, E., Bock, O., Pacione, R., Elgered, G., Vedel, H., Bender, M. Review of the state of the art and future prospects of the ground-based GNSS meteorology in Europe, Atmospheric Measurement Techniques, 9, 5385-5406, doi:10.5194/amt-9-5385-2016, 2016.

Hofmann-Wellenhof, B., Lichtenegger, H., Wasle, E. GNSS – Global Navigation Satellite Systems, Springer, Vienna., Austria, 2008.

Kačmařík, M., Douša, J., Dick, G., Zus, F., Brenot, H., Möller, G., Pottiaux, E., Kapłon, J., Hordyniec, P., Václavovic, P., Morel, L. Inter-technique validation of tropospheric slant total delays, Atmospheric Measurement Techniques, 10, 2183-2208,

doi:10.5194/amt-10-2183-2017, 2017.

Ning, T. a Elgered, G. Trends in the atmospheric water vapour content from ground-based GPS: the impact of the elevation cutoff angle, IEEE Journal of Selected Topics in Applied Earth Observations and Remote Sensing, 5, 744–751, doi:10.1109/JSTARS.2012.2191392, 2012.

Ning, T., Wang, J., Elgered, G., Dick, G., Wickert, J., Bradke, M., Sommer, M., Querel, R., Smale, D. The uncertainty of the atmospheric integrated water vapour estimated from GNSS observations, Atmospheric Measurement Techniques, 9, 1, 79-92, doi:10.5194/amt-9-79-2016, 2016.

Rohm, W.: The ground GNSS tomography – unconstrained approach, Advances in Space Research, 51, 501–513, doi:10.1016/j.asr.2012.09.021, 2013.

Rüeger, J. M. Refractive index formulae for radio waves, FIG XXII International Congress, Washington, D.C. USA, April 19-26, 2002.

Shangguan, M., Bender, M., Ramatschi, M., Dick, G., Wickert, J., Raabe, A., Galas, R. GPS tomography: validation of reconstructed 3-D humidity fields with radiosonde profiles, Annales Geophysicae, 31, 1491-1505, doi:10.5194/angeo-31-1491-2013, 2013.

Solheim, F., Vivekanandan, J., Ware, R., and Rocken, C.: Propagation Delays Induced in GPS Signals by Dry Air, Water Vapor, Hydrometeors, and Other Particulates, J. Geophys. Res., 104, 9663–9670, 1999.

Vey, S., Dietrich, R., Rülke, A., Fritsche, M., Steigenberger, P., Rothacher, M.: Validation of precipitable water vapour within the NCEP/DOE reanalysis using global GPS observations from one decade, Journal of. Climate, 23, 1675–1695, doi:10.1175/2009JCLI2787.1, 2010.

Please also note the supplement to this comment:
https://www.ann-geophys-discuss.net/angeo-2018-76/angeo-2018-76-RC1-

supplement.pdf

---

## Author Comment (AC1) · 20 Sep 2018

To Editor:

Thanks for Editors/Reviewer' warm work earnestly, the corresponding responses to the reviewer's comments are enclosed below. We hope that the corrections meet with approval. Once again, thank you very much for your comments and suggestions.

The following is a point-to-point response to the reviewer's comments.

Response to Referee #1:

The work described in the manuscript is generally worth publishing however the manuscript itself requires major revisions to be accepted - both on the side of content and the way how are the results presented.

**General comments**

1. Based on your findings you provide very strong statements about a superiority of your GNSS results however evidences for it are often either very poor or completely missing. Except radiosonde profiles (which seem to have somehow limited vertical resolution) you do not use any reference product which would support the results provided by your GNSS tomography. You also do not describe at all the meteorological situation itself – what type of precipitation occurred (convective, stratiform), how was it developed, etc. Generally, you directly link the increase of water vapour or its vertical movement to a formation of hydrometeors and consequent rainfall. Although this can be potentially correct, there can be situations where an increase of water vapour or its vertical movement will not lead to any precipitation – just because the rainfall life cycle is not related only to water vapour as it is a much more complex process. Have you checked this? I strongly recommend you to discuss your results with somebody who has strong knowledge in meteorology since I miss this knowledge. It would allow you to much better justify your results. And please see my major comment 6 and 7 for more information regarding this general comment.

✓ Thanks for the reviewer's comments, some information about the tomographic result has been added in Section 4.2. Additionally, the description of meteorological situation during the experimental period has been added in Section 4.

✓ We totally agree with the reviewer's opinion that 'there can be situations where an increase of water vapour or its vertical movement will not lead to any precipitation'. In this manuscript, we analyzed the vertical movement of water vapor during the precipitation period and no-rain period, because we have obtained the hourly precipitation data of 45 rain gauges from the Hong Kong Observatory over the experiment period. We are trying to explain the vertical movement of water vapor during the precipitation period and did not consider the condition that the water vapor increases but without the precipitation.

✓ We appreciate for the reviewer's recommendation, and we have discussed with the people who has strong knowledge in meteorology. Some revisions and suggestions have been added in the manuscript.

2. The overall quality of some of your figures is rather poor and it is really not easy to follow and interpret them. Therefore, I recommend you to do edit some of your figures in the below given major comment 5.

✓ Thanks for the reviewer's suggestion, some figures have been edited and the specific revisions have been given in the response to comment 5.

3. Although the quality of written English is not bad, some issues occur occasionally. I recommend you to let a native speaker proofread your manuscript before a next submission.

✓ Thanks for the reviewer's reminding, the revision manuscript has been proofread by a native English speaker.

**Major comments**

1. I have some comments on how you compute your values of SWV (equation 2,P5L121): - Please use the term horizontal tropospheric gradients instead of just gradients and introduce their meaning.

- It would be worthy to at least mention that horizontal tropospheric gradients represent a gradient of ZTD, not just of ZWD. Are you aware of this? Although during the periods you describe in your study the prevailing gradient was probably the gradient of water vapour, it would be possibly worthy to subtract the hydrostatic part from the total gradient.

- Probably you are aware, that you should use gradient mapping function for mapping the gradients to the elevation angle of the observation, not the wet mapping function (mw). Although your formula corresponds to Bar-Sever gradient mapping function, you should explicitly state it, because there are other gradient mapping functions based on different formulas. In the manuscript you don't mention anywhere which mapping function you did use for hydrostatic and wet components.

- Did you use the conversion factor kappa (which was used to convert ZWD to PWV) also to scale the original values of gradients to "PWV gradients"? I ask because according to your eq. 2 you did not do that and

- Have you considered using post-fit residuals for SWV computation? If not you should at least mention their existence since during severe weather events they can contain important information about tropospheric water vapour distribution which cannot be captured by ZTD or gradients (see i.e. Kacmarík et al., 2017)

✓ Thanks for the reviewer's reminding, the term 'gradients' has been replaced by 'horizontal tropospheric gradients', and their meanings have been introduced.

✓ We are sorry for our improper expression; the horizontal tropospheric gradients represent a gradient of ZWD in our manuscript. Because the hydrostatic gradient has been removed from the total gradient. The corresponding description has been revised in P5L125-127.

✓ Thanks for the reviewer's reminding, the information about which mapping function used in the manuscript has been added in P5L123-124.

✓ Thanks for the reviewer's question, we have used the conversion factor, which has been presented in EQ.1.

✓ Thanks for the reviewer's reminding, to be honest, we haven't used the post-fit residuals in this experiment. The description of post-fit residual for SWV has been added in P5 L127-129.

2. P6L169: Are you sure that solar radiation data provided by a global model with a 0.5x0.5° horizontal resolution and 6h time interval of outputs is a reasonable source of solar radiation data for the level of local meteorological events you work with? And maybe even more importantly: it is absolutely clear that solar radiation is dependent on day/night change and an occurrence of clouds and that it influences the temperature and relative humidity. Why do you include it in your study, what exactly you want to show using it?

✓ Thanks for the reviewer's question, we cannot obtain other solar radiation data currently, therefore, we used solar radiation derived from the CRU-NCEP Ver. 7 dataset. In our opinion, although the spatio-temporal resolution of this dataset is large, it can reflect the general change of solar radiation in our experiment. That's why we select this dataset. This dataset is selected in order to explain the variations of surface temperature and relative humidity, which are influenced by solar radiation.

3. P7L173: I miss important information about your GNSS data processing:
- Which PPP software did you use? Which mode (I guess post-processing), precise products, mapping function, cut-off elevation angle, etc. did you use?
- Are you sure that you estimated ZTD every 30 s (what was then the observation time interval)? Usually while a deterministic modelling of tropospheric parameters is used, the ZTD is estimated in a 5-minute or longer interval (and on P9L232 you also mention that you used PWV in 5-minute interval). Does the software used for your processing is based on a deterministic or a stochastic modelling of troposphere? Since you state that horizontal tropospheric gradients were estimated in 2-hour interval, I guess that it used deterministic approach. Why have you chosen this type of setting to estimate ZTD every 30 s, but gradients only every 2 h?
- What is meant with the presented "accuracy" of estimated ZTD parameters? Is it standard deviation or root-mean-square error or any other statistical parameter? Although you don't provide the information on what exactly these numbers represent, I don't consider 7 or 8 mm as high quality ZTD estimates. For example, the official IGS ZTD product is stated to have an overall accuracy of 4 mm (Byram et al., 2011). I didn't check the referenced paper of Zhao et al. (2018d), however I would suggest you to provide more information on what these values represent and what kind of solutions using Gamit or Bernese were used for these comparisons.

✓ Thanks for the reviewer's question, the important information about GNSS data processing has been added in P7L180-184. We are sorry for our unclear expression which leads to the reviewer's misunderstanding. The ZTD is estimated in every 5 minutes and the sampling rate of GNSS data is 30s. The gradient is estimated in every 2 hours. Because we think the gradient parameter is changed slowly and 2 hours is also the default setting used in GAMIT software, therefore, we selected this value in our experiment.

✓ Thanks for the reviewer's suggestion, the 'accuracy' has been explained and replaced by 'root-mean-square error'. Additionally, the solution model and the representation of those numbers have been added in P7L186-191.

4. P8L208: Could you please explain what presented values of bias/standard deviation mean? Is it a, a comparison between Tm from your regional and standard empirical model or b, a comparison between Tm computed from radiosonde profiles and Tm from your regional and standard empirical model? I guess the b, is right however it is not fully clear. Anyway, I would be careful with your statement that shown results indicate that "the established regional Tm model is superior to the empirical formula". Because if you used radiosonde profiles to correctly establish your regional model, then it MUST be very close to the actual radiosonde profiles. So you only proofed that the established model should provide good results in your area (supposing the radiosonde data are considered to be error-free).

✓ Thanks for the reviewer's reminding, yes, the Tm computed from radiosonde profiles are compared with that from our regional and standard empirical model in the experiment, respectively. This expression has been rewritten in P9L220-223.

5. Comments regarding selected figures:
- Figure 2: since the differences in PWV estimated using different Tm are very small, it is practically impossible to see anything in the figure. Therefore, I propose not to include the figure at all and only optionally provide some statistical information about the variation of PWV based on various Tm values.
- Figure 3, 4, 5, 6: It is really hard to a, see something in detail in these figures, b, compare results in figures 3, 4 with results in figures 5 and 6. In different words, it is really complicated to confirm your written description and interpretation of these figures. Therefore, I strongly recommend you to 1, increase the size of these figures, 2, provide detailed looks on interesting periods (i.e. these with the highest rainfall), 3, try to put all the shown parameters into one figure per station (I mean combine information from figures 3 and 5 and from figures 4 and 6, for example show temperature and Rh together with PWV and rainfall in just one figure – the individual curves can be shifted using a constant offset to increase the readability). I also recommend you to use the same scale in axis y in all figures to make their mutual comparison fair (i.e. in figure 3 you use for PWV interval from 20 to 80 mm, but in figure 4 an interval from 30 to 66 mm).
- Figures 8, 9, 10: I recommend you to provide all these figures as ONE figure with ONE caption. I also recommend you to somehow mark time of a beginning of the precipitation in these figures to increase their readability

✓ Thanks for the reviewer's suggestion, Figure 2 has been removed in the manuscript and some statistical information about the variations of PWV have been given in P9L226-229.
✓ Thanks for the reviewer's suggestion, the size of these figures (3, 4, 5 and 6) has been increased. The scale of axis y in all figures has been unified. We are sorry that we cannot put all parameters into one figure per station because we compare the variables of PWV, RH, and temperature with rainfall separately. If all parameters are presented into one figure, we think this may become more complicated and harder to explain.

✓ Thanks for the reviewer's suggestion, Figures 8, 9 and 10 have been put as one figure with one caption. Additionally, the beginnings of the precipitation in these figures (8, 9 and 10) have been marked with the blue line.

6. Section 4.1 – I miss a reasonable discussion of your results together with their short summary. I can see only a description of your figures together with some generally valid information (like that the solar radiation is connected with day/night cycle or cloud coverage or that the IWV time series can exhibit an increase of values before the beginning of rainfall). What is your interpretation of whole shown time series? Is there any clear relation between PWV and any other meteorological parameter? Are your results in agreement with other researches who studied this topic? Can PWV time series help us to predict rainfall? You should deal with questions like these. Please see also my minor comments related to individual sentences in section 4.1.

✓ Thanks for the reviewer's comments, we are sorry for our unclear expression which leads to the reviewer's confusion. Here, we want to analyze the variations of PWV time series during the rainfall period and no-rainfall period. Additionally, we also analyzed the changes of three meteorological parameter (temperature, relative humidity and solar radiation) during the rainfall period and no-rainfall period. In this section, we did not analyze the relation between PWV and other meteorological parameters. The corresponding content has been rewritten in Section 4.1.

✓ Yes, the PWV time series is capable of forecasting rainfall and the corresponding information has been added in P10L254-255. In fact, we have done some works in this area, the related references are as follows.
References:
Zhao Qingzhi, Yao Yibin and Yao Wanqiang, GPS-based PWV for precipitation forecasting and its application to a typhoon event, Journal of Atmospheric and Solar-Terrestrial Physics, 2018,167:124-133, DOI:10.1016/j.jastp.2017.11.013
Yao Yibin, Shan Lulu and Zhao Qingzhi. Establishing a method of short-term rainfall forecasting based on GNSS-derived PWV and its application, Scientific Reports, 2017,8:12465, DOI:10.1038/s41598-017-12593-z

7. Section 4.2
- P14L306: I would really correct your statement in sentence "In addition, it also can be observed that more sophisticated water vapour variations detected vertically (with 29 layers) can be provided by the GNSS tomographic technique than by radiosonde data." Firstly, looking on figure 7 I consider your radiosonde profiles not to have a full vertical resolution which is standardly accessible by this instrument (standardly you should get one measurement per approximately 30 m). Could you please explain you're your radiosonde profiles are so coarse? Secondly, even if your GNSS tomography could reach a better vertical resolution than a radiosonde (what is not possible with just 29 vertical layers), you could not call your tomography profiles to be "more sophisticated"
– because you are not able to proof that variation in your profiles is related to real meteorological conditions and not only to errors of tomographic reconstruction.

- P18L382: I don't agree with you that "delays to the satellite signals induced by liquid water and icy species : : : are unavailable in the case of GNSS observations". The GNSS signals ARE influenced by them (see Solheim et al., 1999 and Kacmaˇˇrík et al., 2017) and therefore the estimated ZTD/SWD/SIWV contain these effects. However, it is not possible to separate them from the estimated parameters. This also means that your GNSS tomography profiles should be influenced by these hydrometeors.

- I don't see any trustworthy quality evaluation either for your SWV values or for the vertical profiles from GNSS tomography reconstruction. How you can justify that described changes in presented vertical profiles of WV are related to real weather conditions and not only to mismodelling deficiency of your GNSS tomography solution? You should at least discuss this topic.

- Please see also my minor comments related to individual sentences in section 4.2.

- ✓ We appreciate for the reviewer's comments, the expression in P14L306 has been deleted according to the reviewer's suggestion. We have analyzed and removed the data of some layers because of the errors in pressure, water vapor pressure or temperature following some principles (Wang et al., 2008; Wang et al., 2016). Therefore, the radiosonde profiles are relatively coarse. Additionally, we agree with the reviewer's opinion that we cannot call the tomographic profiles to be 'more sophisticated', therefore, this expression has been removed from the manuscript. We don't know where we can download the radiosonde data with the vertical resolution of about 30 m. We have checked the radiosonde files carefully, which downloaded from ftp://ftp.ncdc.noaa.gov/pub/data/igra/ and found that the vertical resolution is non-uniformed with the value ranges from several hundred meters to more than one kilometer.

- ✓ Thanks for the reviewer's opinion. We are sorry for our unclear expression which leads to the reviewer's misunderstanding. In our opinion, the GNSS tomography profiles are influenced by these hydrometeors, but not evident like the delays induced by atmospheric water vapor. In other words, the satellite signals induced by liquid water and icy species are very small (Brenot et al., 2014), which cannot be reflected by GNSS observation evidently. This improper expression has been rewritten in P18L387-389.

- ✓ Thanks for the reviewer's suggestion, the comparison experiment of tomographic result with radiosonde data has been performed and presented in the first paragraph of Section 4.2. Due to this paper is mainly focus on the analysis of 2-d-/4-d water vapour variation during the heavy rainfall events and the quality evaluations of vertical profiles from GNSS tomography reconstruction have been carried out in this research area in the previously studies (Yao and Zhao, 2016, 2017; Zhao et al., 2017; Zhao et al., 2018). Therefore, we didn't give much information in this aspect. Some information corresponding the quality evaluation of tomographic result have been added in P13L317-321.

References:

Wang, J., & Zhang, L. (2008). Systematic errors in global radiosonde precipitable water data from comparisons with ground-based gps measurements. J Clim, 21(10), 2218-2238.

Wang, X., Zhang, K., Wu, S., Fan, S., & Cheng, Y. (2016). Water vapor‑weighted mean temperature and its impact on the determination of precipitable water vapor and its linear trend. Journal of Geophysical Research Atmospheres, 121(2).

Brenot, H., Wautelet, G., Warnant, R., Neméghaire, J., & Roozendael, M. V. (2014). GNSS meteorology and impact on NRT position. Enc-Gnss.

Yao Y.; Zhao Q. Maximally Using GPS Observation for Water Vapor Tomography. IEEE Transactions on Geoscience and Remote Sensing, 2016, 54(12), 7185-7196.

Yao Y.; Zhao Q. A novel, optimized approach of voxel division for water vapor tomography[J]. Meteorology and Atmospheric Physics, 2017, 129(1), 57-70.

Zhao Q.; Yao Y.; Yao W. A troposphere tomography method considering the weighting of input information[C]//Annales Geophysicae. Copernicus GmbH, 2017, 35(6), 1327-1340.

Zhao Q., Yao Y., Cao X., Zhou F. and Xia P. An Optimal Tropospheric Tomography Method Based on the Multi-GNSS Observations. Remote Sensing, 2018, 10(2), 234.

**Minor comments**

1. P2L42: you write about using GNSS PWV for severe weather and climate studies, however you provide only two references. Since there are many studies related to GNSS meteorology and (a) severe weather monitoring (i.e. Japanese team around Y. Shoji); (b) climate (i.e. Gradinarsky et al., 2002, Vey et al., 2010, Ning a Elgered 2012, Bock et al., 2014), I recommend you to cite at least the most important ones and cite them separately for (a) and (b).

✓ Thanks for the reviewer's reminding, this expression has been revised and the corresponding references have been added separately in P2L43-45.

References:

Shoji, Y. (2009). Assimilation of nationwide and global gps pwv data for a heavy rain event on 28 july 2008 in hokuriku and kinki, japan. SOLA - Scientific Online Letters on the Atmosphere, 5(1), 45-48.

Shoji, Y. (2013). Retrieval of water vapor inhomogeneity using the japanese nationwide gps array and its potential for prediction of convective precipitation. Journal of the Meteorological Society of Japan, 91(1), 43-62.

Gradinarsky, L. P., Johansson, J., Bouma, H. R., Scherneck, H. G., Elgered, G.Climate monitoring using GPS, Physics and Chemistry of the Earth, 27, 335–340, doi:10.1016/S1474-7065(02)00009-8, 2002.

Vey, S., Dietrich, R., Rülke, A., Fritsche, M., Steigenberger, P., Rothacher, M.: Validation of precipitable water vapour within the NCEP/DOE reanalysis using global GPS observations from one decade, Journal of. Climate, 23, 1675–1695, doi:10.1175/2009JCLI2787.1, 2010.

Ning, T. a Elgered, G. Trends in the atmospheric water vapour content from ground-based GPS: the impact of the elevation cutoff angle, IEEE Journal of Selected Topics in Applied Earth Observations and Remote Sensing, 5, 744–751, doi:10.1109/JSTARS.2012.2191392, 2012.

Bock, O., Willis, P., WaDng, J., Mears, C. A high-quality, homogenized, global,

long-term (1993–2008) DORIS precipitable water data set for climate monitoring and model verification, Journal of Geophysical Research, 119, 7209–7230, doi:10.1002/2013JD021124, 2014

2. P2L44: what do you mean with "traditional sounding stations" – radiosondes? It should be explicitly given. Are you aware of WV observations from remote sensing satellites or of other instruments as WVR or Raman Lidar?

✓ Thanks for the reviewer's reminding, "traditional sounding stations" refers to radiosondes. This expression has been revised in P2L44. Yes, the atmospheric water vapor information also can be obtained based on the remote sensing technique or Raman Lidar.

3. P2L50: I would rather write that GNSS PWV can be used to provide information about water vapour distribution, which is related to form of precipitation and not only to severe weather events. In this regard – how do you define a severe weather event in your perspective?

✓ Thanks for the reviewer's suggestion, this expression has been revised in P2L53-54.

✓ In our opinion, the severe weather event mentioned in this manuscript mainly refer to the severe precipitation or some events related to this.

4. P2L52: I would rather write that GNSS PWV is operationally used for their assimilation into numerical weather prediction models (NWM) than just for operational meteorology. Italy doesn't use GNSS PWV operationally, work presented in Barindelli et al. (2018) was just a case study to promote an operational deployment.

✓ We appreciate for the reviewer's comments earnestly and we are sorry for our improper expression about the work presented in Barindelli et al. (2018). This expression has been revised according to the reviewer's suggestion in P2L55-57.

5. P2L56: I would rather say that ZTD or PWV can be used for early warnings than that it is used. I recommend you here to cite the work of Brenot et al. (2013) at least.

✓ Thanks for the reviewer's suggestion, this sentence has been revised and the work of Brenot et al. (2013) has been added P1L59-60.

6. P3L65: there are much more (recent) studies that used radiosonde profiles for GNSS tomography validations, i.e. Shangguan et al. (2013).

✓ Thanks for the reviewer's reminding, the corresponding reference has been added in P2 L69.

7. P3L68: COSMIC is not an instrument for water vapour sensing, it is whole program designed for various purposes – in this regard I would rather write GNSS Radio occultation technique to be consistent.

✓ Thanks for the reviewer's reminding, the improper expression has been revised in P2 L71.

8. P3L75: I would not say that iterative reconstruction techniques deal with resolution of tomography models or division of tomography areas.

✓ Thanks for the reviewer's reminding, this expression has been removed in the manuscript.

9. P4L91: please correct your statement that ionosphere causes signal delay – since ionosphere causes a delay for code measurements, but an advance for phase measurements (see i.e. Hofmann-Wellenhof et al. 2008). Also you cannot fully eliminate ionosphere with just the IF linear combination – this eliminates only the first order effect but not effects of higher order.

✓ Thanks for the reviewer's suggestion, this expression has been rewritten in P4L93-94.

10. P4L95: I would firstly introduce the hydrostatic and non-hydrostatic part of the delay and then call the non-hydrostatic the wet delay. Also please use zenith direction all the time (instead of a vertical direction).

✓ Thanks for the reviewer's suggestion, this expression has been rewritten according to the reviewer's suggestion in P4L95-97. Additionally, the 'vertical direction' has been replaced by 'zenith direction' throughout the manuscript.

11. P4L101: replace the word measurements with a more appropriate signal.

✓ Thanks for the reviewer's suggestion, this word has been replaced by 'observations'.

12. P4L102: provide references for given processing software.

✓ Thanks for the reviewer's reminding, the corresponding references have been added.

13. P4L103: you should rather mention that in the GNSS data processing the ZHD is usually taken from an a priori model and later precisely computed from i.e. real meteorological observations to subtract ZWD from ZTD. Because strictly speaking in the GNSS data processing you do not estimate a ZWD, but a correction to a priori modelled ZHD.

✓ Thanks for the reviewer's suggestion, this expression has been added in P4L101-103.

14. P4L107: use rather base SI units for all the given coefficients. It would be also worthy to mention that different sets of refractivity constants exist (see Rueger, 2002).

✓ Thanks for the reviewer's suggestion and reminding, the SI units for all given coefficients have been given in P4L108.

15. P5L117: I recommend replacing the term SWV with SIWV (Slant integrated water vapour).

✓ Thanks for the reviewer's suggestion, the SWV has been replaced by SIWV throughout the manuscript.

16. P5L142: Rohm (2013) developed a GNSS tomography solution using no constraints, so please correct your sentence according to that.

✓ Thanks for the reviewer's reminding, this expression has been corrected in P5L147-148.

17. P6L169: could you provide a reference for given CRU-NCEP solar radiation data sets you used in your study?

✓ Thanks for the reviewer's suggestion, the corresponding reference has been added in P6L176.

18. P7L191: with the "layered parameters" you mean that information about vertical profile of some meteorological parameters as water vapour pressure or air temperature is not available? I recommend you rewriting this sentence to make better understandable what you mean.

✓ Thanks for the reviewer's suggestion, this expression has been rewritten to make it better understandable in P8L204-205.

19. P8L209: I recommend you to see the paper of Ning et al. (2016) providing a rigorous evaluation of uncertainty in GNSS IWV estimation including impact of Tm uncertainty. I would also like to put you into perspective with the number of overall achievable accuracy of IWV estimates from GNSS which is meant to be around 0.4

– 0.6 mm (see Guerova et al., 2016) – it would be possible good to mention it in the paper.

✓ We appreciate for the reviewer's suggestion and comments, we have learned the paper and cited the result obtained by Guerova et al. (2016) in the manuscript in P9L232-233.

20. P9L227: 300 mm is a value for one day and one station? Or for whole studied period from July 19 till July 25? It is not clear from your sentence.

✓ Thanks for the reviewer's reminding, this expression has been rewritten in P9L237-239 to make it clear.

21. P9L240: with the sentence "Additionally, the PWV time series data present a downward trend at four stations during this period" you mean the whole processed period from June 19 till July 25 or anything else? If yes, why do you state it? It is from whatever reason important for your study?

✓ Thanks for the reviewer's question, yes, the variation of PWV time series shows a downward trend during the experimental period. This expression is not important for our study; therefore, we delete this expression in the manuscript.

22. P10L244: please add information at which stations these values of cumulative rainfall occurred.
   ✓ Thanks for the reviewer's reminding, the name of stations have been added in the manuscript.

23. P10L249: what do you want to say with your sentence that "the PWV values during rainfall are much larger than that of no rainfall time" You just want to state the fact, that this situation occurred in your selected time periods or you want to state it as a general fact which is valid every time? Because I would not agree with the second option.
   ✓ Here, we want to express that the PWV values during rainfall are much larger than that of no rainfall time at the situation occurred in the selected time period. This expression has been revised according to the reviewer's comments.
   ✓ Additionally, we totally agree with the reviewer's opinion that the PWV values are not often larger during rainfall period than that of no rainfall period.

24. P14L314: The sentence starting "For the SPP rain gauge" is not understandable for me.
   ✓ Thanks for the reviewer's reminding, this sentence has been revised to make it understandable.

25. P14L325: I would consider rewriting the sentence starting "The above phenomenon indicates". The heavy rainfall could be induced by vertical motion of water vapour, but the opposite (variation of water vapour induced by heavy rainfall) is not so logical.
   ✓ Thanks for the reviewer's suggestion, this expression has been revised in P14L339-340.

26. Table 1 and Table 2: could you provide a more detailed look on the precipitation? I mean provide the rainfall information not only in 1-hour interval, but for example in 5-minutes or 10-minutes time interval. You don't need to provide it in a table, in can be in a figure (optionally in the figures 8, 9, 10 themselves or in figure 11). It would provide the reader a much better idea how was the rain spread out in time.
   ✓ Thanks for the reviewer's suggestion, we are sorry that we cannot provide the rainfall information with a higher temporal resolution. Currently, we only bought the hourly rainfall data of 45 rain gauges from the Hong Kong Observatory for the experimental period.

We appreciate for Reviewer's warm work earnestly, which has a significant improvement for our manuscript. And we hope that our corrections meet with the reviewer's requirement. Once again, thank you very much for your comments and suggestions.

---

## Author Comment (AC2) · 20 Sep 2018

[revised manuscript text omitted]

**2 Fundamentals of GNSS meteorology**

**2.1 Retrieval of GNSS PWV**

Satellite signals are delayed and bent when crossing the troposphere, which can be divided into two parts: hydrostatic delay and non-hydrostatic delay. The first part in a zenith direction, also called zenith hydrostatic delay (ZHD), can be precisely calculated by the Saastamoinen model (Saastamoinen, 1972) with the observed surface pressure. The second part can be estimated in the zenith direction using GNSS data, which is also called zenith wet delay (ZWD), from which the PWV can be calculated. Therefore, GNSS meteorology is formed, as first proposed by Bevis et al. (1992). The calculation used in obtaining PWV is expressed as follows: the zenith total delay is first estimated by processing the GNSS observations using the GNSS processing software such as Bernese (Dach et al., 2015), GAMIT (Herring et al., 2010), *etc*. In the GNSS data processing, the ZHD is usually taken from an a priori model and later precisely computed from i.e. real meteorological observation. The ZWD is then obtained by extracting the ZHD from ZTD and thus the PWV can be calculated based on the following equations (Saastamoinen, 1972; Askne and Nordius, 1987; Bevis et al., 1992):

$$
\begin{aligned}
&\text{PWV} = \Pi \cdot \text{ZWD} \\
&\Pi = 10^6 \Big/ \left( (k_2' + k_3 / \text{Tm}) \cdot R_v \cdot \rho_w \right) \\
&\text{ZWD} = \text{ZTD-ZHD} \\
&\text{ZHD} = \frac{0.002277 \times \text{P}}{1 - 0.00266 \times \cos(2\varphi) - 0.00028 \times \text{H}}
\end{aligned}
\tag{1}
$$

Where $\Pi$ refers to the conversion factor. $k_2'$, $k_3$, and $R_v$ are constants with values of 22.1 K/hPa, $3.739 \times 10^5$ K$^2$/hPa and 461.495 J/kg/K, respectively. $T_m$ represents the weighted mean temperature, which is related to surface parameters such as temperature and pressure. Therefore, $T_m$ is usually calculated based on the empirical model using the data from radiosonde or numerical weather model due to the observed layered meteorological parameters which are rarely obtained (Bevis et al., 1994; Yao et al., 2012). In the fourth formula in Eq. (1), P, H and $\varphi$ represent the surface pressure (hPa), geodetic height (km) and station latitude (rad), respectively. In our study, the value of $T_m$ is calculated based on the established regional $T_m$ model using the radiosonde data and observed temperature (Section 3.2).

**2.2 Establishment of tomographic model**

Generally, the slant wet delay (SWD) or slant integrated water vapour (SIWV) is considered as the input information for GNSS troposphere tomography (Flores et al., 2000; Hirahara, 2000; Skone and Hoyle, 2005; Rohm and Bosy, 2009; Chen and Liu., 2014) and the following equation gives an expression used to obtain SIWV (Flores et al., 2000):

$$\text{SIWV}_{azi,ele} = m_w(ele) \cdot \text{PWV} + m_w(ele) \cdot cot(ele) \cdot (G_{NS}^w \cdot cos(azi) + G_{WE}^w \cdot sin(azi)) \tag{2}$$

Where $m_w$ represents the gradient mapping function, and global mapping function (GMF) is used in our experiment (Böhm et al., 2006). *ele* and *azi* refer to the elevation angle and azimuth angle, respectively. $G_{NS}^w$ and $G_{WE}^w$ are the horizontal tropospheric gradient parameters of ZWD in the south-north and west-east directions, respectively, which are caused by the atmospheric inhomogeneous. Additionally, a post-fit residual for SIWV is existed, which contains some information about tropospheric water vapor distribution during the severe weather events and cannot captured by ZTD or gradients (Kacmarík et al., 2017).

If a sufficient number of SIWVs derived from some stations in a regional CORS network can be obtained, the GNSS tomographic technique can be used to reconstruct the three-dimensional (3-d)

distribution of atmospheric water vapour field. Therefore, a four-dimensional (4-d) water vapour information is a time series of such a 3-d tomographic result, which can be used to reflect the regional atmospheric water vapour variations in both the spatial and temporal domains. As described by Flores et al. (2000), the linear observation equation between SIWV and water vapour density can be expressed as follows:

$$\text{SIWV} = \sum (d_{ijk} \cdot x_{ijk}) \tag{3}$$

Where $i, j$ and $k$ represent the location of the area of interest in the longitudinal, latitudinal, and zenith directions, respectively. $d_{ijk}$ and $x_{ijk}$ refer to the distance travelled by satellite signals and the water vapour density remains to be estimated, respectively in the discretized voxels ($i, j, k$). Therefore, the matrix form of the tomographic observation equation can be described as follows:

$$\mathbf{y} = \mathbf{A} \cdot \mathbf{x} \tag{4}$$

Where $\mathbf{y}$ represents the column vector of SIWV derived from GNSS measurements. $\mathbf{A}$ and

$\mathbf{x}$ are the coefficient matrix of distance penetrated by satellite rays and the column vector of water vapour density, respectively.

Although Rohm (2013) developed a GNSS tomography solution using no constraints, most studies still used some constraints to overcome the influence caused by the ill-posed problem in the inversion of the tomographic normal equation (Flores et al., 2000; Bi et al., 2006; Bender et al.,

2011; Rohm and Bosy, 2011 Chen and Liu, 2014). In our study, both horizontal and vertical constraints are considered. The water vapour density in a certain voxel is regarded as the weighted mean value of its horizontal neighbouring voxels (Rius et al., 1997) and the negative exponential function is introduced to describe the relationship between the nearby voxels in the zenith direction while the coefficients of functional model are established using radiosonde data (Yao and

Zhao, 2016). Consequently, the tomographic modelling can be expressed after imposing the constraints as:

$$\begin{pmatrix} \mathbf{y} \\ \mathbf{0} \\ \mathbf{0} \end{pmatrix} = \begin{pmatrix} \mathbf{A} \\ \mathbf{H} \\ \mathbf{V} \end{pmatrix} \cdot \mathbf{x} \qquad (5)$$

Where $\mathbf{H}$ and $\mathbf{V}$ are the coefficient matrices of horizontal and vertical equations, respectively.

To obtain a reasonable tomographic result from the above equation, an optimal tropospheric solution method is used, which can adaptively tune the weightings of different types of equations (Zhao et al., 2018d).

**3 Data description and establishment of a regional $T_m$ model**

**3.1 Data description**

To validate the ability of GNSS technique in capturing the signature of atmospheric water vapour variation during heavy rainfall events, two periods of GNSS observations (19 to 27, July 2015 and

1 to 8, August 2015) from 13 GNSS stations in the CORS network of Hong Kong are selected in the experiment. Those two periods are selected because they correspond to a heavy rainfall event and a no-rainfall event, respectively according to hourly rainfall data from 45 rain gauges evenly distributed across this area (Figure 1). There is a radiosonde station located in this area where the radiosonde balloon is launched twice daily at UTC 00:00 and 12:00, respectively. The 20-years of radiosonde data from 1998 to 2017 are used to establish the regional $T_m$ model in this study. In addition, the surface temperature and relative humidity are also selected to analyse their changes during those two periods. To explain the variations of surface temperature and relative humidity, the solar radiation data are also used in this study, which is derived from the CRU-NCEP Ver. 7 dataset (Wu et al., 2015). This dataset is a combination product of the CRU TS3.2 climate dataset and the NCEP reanalysis data. The temporal-spatial resolution of the solar radiation dataset are four times daily (UTC 00:00, 06:00, 12:00 and 18:00) and 0.5°×0.5°, respectively.

GNSS observations are processed using Precise Point Positioning (PPP) data processing software developed by our research group. The post-processing mode is used and the orbit and satellite clock errors are corrected using the final products downloaded from ftp://ftp.gfz-potsdam.de/. The GMF is selected (Böhm et al., 2006) while the cut-off elevation angle is 7° in our experiment. The sampling rate of GPS observation is 30s and the and the ZTD parameter is estimated every 5 minutes. The gradient parameters in south-north and east-west directions are also estimated at intervals of 2 h. The detailed description of processing strategy has been presented in Zhao et al. (2018d). The root-mean-square error (RMSE) of the estimated ZTD parameter has been proved with the values of 7.2 mm and 8.1 mm when the ZTD estimated from the GAMIT (v10.5) and Bernese (v5.2) software based on the double-difference model are regarded as references, respectively (Zhao et al., 2018a). Due to the accurately ZHD parameter can be estimated using the empirical model, therefore, the final error in the estimated PWV values is approximately 1-1.5mm (Zhao et al., 2018e). The corresponding meteorological parameters, such as the surface pressure and temperature, are also obtained at the selected GNSS stations. Therefore, the precise ZHD can be calculated by the empirical model using the observed surface pressure. The conversion factor, as described in Eq. (1), is also obtained, in which $T_m$ is calculated based on the established $T_m$ model which will be introduced in the following section. Finally, the PWV time series, as well as the SIWVs for the 13 selected GNSS stations, can be obtained. Five of the 45 rain gauges (R21, TMS, PEN, SSP, and KSC) are selected to analyse the variations in atmospheric water vapour during different weather conditions (Figure 1).

[Figure]

Figure 1. Geographic distribution of selected GNSS and radiosonde stations as well as the rain gauges used in the experiment

**3.2 Establishment of the regional $T_m$ model**

Due to the layered information about vertical profile of some meteorological parameters as water vapour pressure or air temperature is generally unavailable for the location of GNSS stations, therefore, the $T_m$ values of those stations are calculated based on the empirical model in this experiment. It has been proved that $T_m$ is highly correlated with the variations of temperature, pressure, and the seasons (Bevis et al., 1992; Yao et al., 2012; Yao et al., 2014, 2015; Liu et al.,

2018). Therefore, a regional $T_m$ model which includes as parameters: temperature, surface pressure, and seasonal variation, is established and expressed as follows:

$$T_m = T_{m0} + a*T_s + b*P_s + c*\cos(2\pi\frac{doy}{365.25}) + d*\sin(2\pi\frac{doy}{365.25})$$
$$+e*\cos(4\pi\frac{doy}{365.25}) + f*\sin(4\pi\frac{doy}{365.25})$$

(6)

Where $T_{m0}$, $T_s$ and $P_s$ represent the initial value of $T_m$, surface temperature and surface pressure, respectively; $doy$ refers to the day of year; $a$ and $b$ are coefficients of $T_s$ and $P_s$, respectively, while $c$ to $f$ refer to the coefficients of the seasonal correction function. In our study, the coefficients in Eq. (6) were obtained by the least square regression method using

20-year radiosonde data for 45004 while the values of $a$ to $f$ are 129.1225, 0.5370, -0.0023,

0.358, 0.813, -0.178, and 0.255, respectively.

The performance of the established $T_m$ model is analysed and compared with the empirical formula proposed by Bevis et al. (1994). Statistical result of 20-years of radiosonde data reveals that when compared to the $T_m$ values calculated using the observed radiosonde data, the standard deviation and bias for the established $T_m$ model and the empirical formula proposed by Bevis et al. (1994) are 2.04/0.0009 K and 3.41/2.53 K, respectively, which indicates that the established regional $T_m$ model shows a good result in research area. To further analyse the impact of $T_m$ model error on the calculated PWV, a comparison experiment is carried out for radiosonde station 45004 with a variation in $T_m$ of 1 K, 3 K, 5 K, 7 K, and 9 K, respectively and compared with the actual PWV values over the period of 1998 to 2017. Statistical analysis shows that the PWV errors induced by the change in $T_m$ of 1 K, 3 K, 5 K, 7 K, and 9 K are 0.15 mm, 0.45 mm, 0.75 mm, 1.04 mm, and 1.34 mm, respectively under the condition of PWV > 0 mm, while the values are 0.18 mm, 0.54 mm, 0.91 mm, 1.27 mm, and 1.63 mm, respectively when PWV > 40 mm. Therefore, the PWV errors caused by the established $T_m$ model in this study are less than 0.4 mm and 0.5 mm when PWV > 0 mm and PWV > 40 mm, respectively. Such result is deemed acceptable for the analysis of PWV variations with rainfall events (Akilan et al., 2015) and corresponds to the previous result obtained from Guerova et al. (2016).

**4 Signature of 2-d/4-d variations in atmospheric water vapour during rainfall**

According to the recordings derived from the Hong Kong Observatory, the convectional rain happened during the period of 19 to 27, July 2015. It is continuous rains in Hong Kong and the hourly rainfall have been accumulated for 45 rain gauges with the largest rainfall more than 300 mm for the entire experimental period in HKSC station. The weather conditions are cloudy and sunny without rainfall happened for the period of 1 to 8, August 2015. Therefore, those two periods are selected in this paper to investigate the variation characteristics of atmospheric water vapor.

**4.1 Cases of 2-d PWV time series change**

To capture the signature of PWV time series change in different weather conditions, the comparison between the 5-minute GNSS-derived PWV and hourly rainfall are performed for the periods of 19 to 27, July 2015 and 1 to 8, August 2015, respectively. Four GNSS stations (HKKS, HKSC, HKPC, and HKSL) and the surrounding rainfall gauges (HSC, SSP, PEN, and R21) are selected for this experiment. Additionally, other meteorological parameters (temperature, relative humidity and solar radiation) are also analysed during this experimental period.

Figure 2 shows the variations of 5-minute PWV time series data with hourly rainfall as well as the cumulative rainfall at those four stations for the period of 19 to 27, July 2015. It can be seen from the first column of Figure 2 that the PWV time series show an increasing trend before the arrival of rainfall and reaches a relatively large value during rainfall, PWV then returns to its average value after rainfall. Such phenomenon found above can be used to forecast the nowcasting rainfall (Yao et al., 2017; Zhao et al., 2018a). The second column of Figure 2 reveal that the cumulative rainfall first increased at about UTC 11:00, 20 July, 2015 with different levels reached and the event terminated at UTC 12:00, 23 July, 2015. The largest cumulative rainfall reached more than

300 mm at HKSC station while the minimum recorded rainfall was about 100 mm at HKSL

[revised manuscript text omitted]

Bock, O., Willis, P., WaDng, J., Mears, C. A high-quality, homogenized, global, long-term (1993–2008) DORIS precipitable water data set for climate monitoring and model verification, Journal of Geophysical Research, 119, 7209–7230, doi:10.1002/2013JD021124, 2014

Böhm, J., Niell, A., Tregoning, P., & Schuh, H. (2006). Global Mapping Function (GMF): A new empirical mapping function based on numerical weather model data. Geophysical Research Letters, 33(7).

Braun, John Joseph. Remote sensing of atmospheric water vapor with the global positioning system. Geophysical Research Letters, 2004, 20(23), 2631-2634.

Braun J.; Rocken C.; Liljegren J. Comparisons of line-of-sight water vapor observations using the global positioning system and a pointing microwave radiometer. *Journal of Atmospheric and Oceanic Technology*, 2003, 20(5), 606-612.

Brenot H.; Neméghaire J.; Delobbe L.; et al. Preliminary signs of the initiation of deep convection by GNSS. *Atmospheric Chemistry and Physics*, 2013, 13(11), 5425-5449.

Brenot, H., Ducrocq, V., Walpersdorf, A., Champollion, C., & Caumont, O. (2006). GPS zenith delay sensitivity evaluated from high‐resolution numerical weather prediction simulations of the 8–9 September 2002 flash flood over southeastern France. *Journal of Geophysical Research: Atmospheres*, 111(D15).

Chen B, Liu Z. Voxel-optimized regional water vapor tomography and comparison with radiosonde and numerical weather model. *Journal of geodesy*, 2014, 88(7), 691-703.

Dach, R., Lutz, S., Walser, P., Fridez, P., 2015. Bernese GNSS Software Version 5.2. Astronomical Institute, University of Bern, 884.

Flores A.; Ruffini G.; Rius A. 4D tropospheric tomography using GPS slant wet delays//Annales Geophysicae. *Springer-Verlag*, 2000, 18(2), 223-234.

Gradinarsky, L. P., Johansson, J., Bouma, H. R., Scherneck, H. G., Elgered, G.Climate monitoring using GPS,

Physics and Chemistry of the Earth, 27, 335–340, doi:10.1016/S1474-7065(02)00009-8, 2002.

Guerova G.; Jones J.; Dousa J.; et al. Review of the state of the art and future prospects of the ground-based GNSS meteorology in Europe. *Atmospheric Measurement Techniques*, 2016, 9(11), 1-34.

Heublein M.; Zhu X X.; Alshawaf F.; et al. Compressive sensing for neutrospheric water vapor tomography using GNSS and InSAR Observations[C]//Geoscience and Remote Sensing Symposium (IGARSS), *2015 IEEE International*. IEEE, 2015, 5268-5271.

Herring, T.A., King, R.W., McClusky, S.C., 2010. Documentation of the GAMIT GPS Analysis Software Release 10.4. Department of Earth and Planetary Sciences, Massachusetts Institute of Technology, Cambridge, Massachusetts.

Hirahara K. Local GPS tropospheric tomography. *Earth, planets and space*, 2000, 52(11), 935-939.

JMA (2013) Outline of the operational numerical weather prediction at the Japan Meteorological Agency. Appendix to WMO technical progress report on the global data-processing and forecasting system (GDPFS) and numerical weather prediction (NWP) research. http://www.jma.go.jp/jma/jma-eng/jma-center/nwp/outline2013-nwp/index.htm. Accessed 30 Aug 2017

Kacmarík, M., Douša, J., Dick, G., Zus, F., Brenot, H., Möller, G., Pottiaux, E., Kapłon, J., Hordyniec, P., Václavovic, P., Morel, L. Inter-technique validation of tropospheric slant total delays, Atmospheric Measurement Techniques, 10, 2183-2208, doi:10.5194/amt-10-2183-2017, 2017

Liu J.; Yao Y.; Sang J. A new weighted mean temperature model in China. *Advances in Space Research*, 2018, 61(1), 402-412.

Liu Z.; Wong M S.; Nichol J.; et al. A multi-sensor study of water vapour from radiosonde, MODIS and AERONET: a case study of Hong Kong. *International Journal of Climatology*, 2013, 33(1), 109-120.

Ning, T. a Elgered, G. Trends in the atmospheric water vapour content from ground-based GPS: the impact of the elevation cutoff angle, IEEE Journal of Selected Topics in Applied Earth Observations and Remote Sensing, 5, 744–751, doi:10.1109/JSTARS.2012.2191392, 2012.

Perler D.; Geiger A.; Hurter F. 4D GPS water vapor tomography: new parameterized approaches. Journal of Geodesy, 2011, 85(8), 539-550.

Rius A.; Ruffini G.; and Cucurull L.; "Improving the vertical resolution of ionospheric tomography with GPS occultations," *Geophys*. Res. Lett. 1997, 24(18), 291–2294.

Rohm, W.: The ground GNSS tomography – unconstrained approach, Advances in Space Research, 51, 501–513, doi:10.1016/j.asr.2012.09.021, 2013

Rohm W, Bosy J. Local tomography troposphere model over mountains area. *Atmospheric Research*, 2009, 93(4), 777-783.

Rohm W.; Bosy J. The verification of GNSS tropospheric tomography model in a mountainous area . *Advances in Space Research*, 2011, 47(10), 1721-1730.

Saastamoinen, J. Atmospheric correction for the troposphere and stratosphere in radio ranging satellites. *The use of artificial satellites for geodesy*, 1972, 247-251.

Saito K.; Shoji Y.; Origuchi S.; et al. GPS PWV Assimilation with the JMA Nonhydrostatic 4DVAR and Cloud Resolving Ensemble Forecast for the 2008 August Tokyo Metropolitan Area Local Heavy Rainfalls//*Data Assimilation for Atmospheric, Oceanic and Hydrologic Applications (Vol. III)*. Springer, Cham, 2017, 383-404.

Shoji, Y. (2009). Assimilation of nationwide and global gps pwv data for a heavy rain event on 28 july 2008 in hokuriku and kinki, japan. SOLA - Scientific Online Letters on the Atmosphere, 5(1), 45-48.

Shoji, Y. (2013). Retrieval of water vapor inhomogeneity using the japanese nationwide gps array and its potential for prediction of convective precipitation. Journal of the Meteorological Society of Japan, 91(1), 43-62.

Shangguan, M., Bender, M., Ramatschi, M., Dick, G., Wickert, J., Raabe, A., Galas, R. GPS tomography: validation of reconstructed 3-D humidity fields with radiosonde profiles, Annales Geophysicae,

31, 1491-1505, doi:10.5194/angeo-31-1491-2013, 2013.

Skone S.; Hoyle V. Troposphere Modeling in a Regional GPS Network. *Positioning*, 2005, 4(1&2), 230-239.

Troller M.; Geiger A.; Brockmann E.; et al. Determination of the spatial and temporal variation of tropospheric water vapour using CGPS networks. *Geophysical Journal International*, 2006, 167(2), 509-520.

Vey, S., Dietrich, R., Rülke, A., Fritsche, M., Steigenberger, P., Rothacher, M.: Validation of precipitable water vapour within the NCEP/DOE reanalysis using global GPS observations from one decade, Journal of. Climate, 23, 1675–1695, doi:10.1175/2009JCLI2787.1, 2010.

Wang, X., Wang, X., Dai, Z., et al. Tropospheric wet refractivity tomography based on the BeiDou satellite system. *Advances in Atmospheric Sciences*, 2014, 31(2), 355-362.

Wu, D., Zhao, X., Liang, S., Zhou, T., Huang, K., Tang, B., and Zhao W. (2015). Time-lag effects of global vegetation responses to climate change. Global Change Biology, 21(9), 3520-3531.

Yao Y B.; Zhu S.; Yue S Q. A globally applicable, season-specific model for estimating the weighted mean temperature of the atmosphere. *Journal of Geodesy*, 2012, 86(12), 1125-1135.

Yao Y B.; Zhao Q Z.; Zhang B. A method to improve the utilization of GNSS observation for water vapor tomography. *Annales Geophysicae (09927689)*, 2016, 34(1), 143-152.

Yao Y.; Zhao Q. Maximally Using GPS Observation for Water Vapor Tomography. *IEEE Transactions on Geoscience and Remote Sensing*, 2016, 54(12), 7185-7196.

Yao Y.; Zhao Q. A novel, optimized approach of voxel division for water vapor tomography[J]. *Meteorology and Atmospheric Physics*, 2017, 129(1), 57-70.

Yao Y.; Shan L.; Zhao Q. Establishing a method of short-term rainfall forecasting based on GNSS-derived PWV and its application. *Scientific reports*, 2017, 7(1), 12465.

Yao Y.; Zhang B.; Xu C. and Yan, F. Improved one/multi-parameter models that consider seasonal and geographic variations for estimating weighted mean temperature in ground-based GPS meteorology. *Journal of Geodesy*, 2014, 88(3), 273-282.

Yao Y B.; Liu J H.; Zhang B.; et al. Nonlinear relationships between the surface temperature and the weighted mean temperature. *Geomatics & Information Science of Wuhan University*, 2015, 40(1), 112-116.

Zhang K.; Manning T.; Wu S.; et al. Capturing the signature of severe weather events in Australia using GPS measurements. *IEEE Journal of Selected Topics in Applied Earth Observations and Remote Sensing*, 2015, 8(4), 1839-1847.

Zhao Q.; Yao Y. An improved troposphere tomographic approach considering the signals coming from the side face of the tomographic area//Annales Geophysicae. *Copernicus GmbH*, 2017b, 35(1), 87-95.

Zhao Q.; Yao Y.; Cao X.; et al. Accuracy and reliability of tropospheric wet refractivity tomography with GPS, BDS, and GLONASS observations. *Advances in Space Research*, 2018c. DOI:10.1016/j.asr.2018.01.021

Zhao Q.; Yao Y.; Yao W. A troposphere tomography method considering the weighting of input information[C]//Annales Geophysicae. *Copernicus GmbH*, 2017a, 35(6), 1327-1340.

Zhao Q.; Yao Y.; Yao W. GPS-based PWV for precipitation forecasting and its application to a typhoon event. *Journal of Atmospheric and Solar-Terrestrial Physics*, 2018a, 167, 124-133.

Zhao Q., Yao Y., Cao X., Zhou F. and Xia P. An Optimal Tropospheric Tomography Method Based on the Multi-GNSS Observations. *Remote Sensing*, 2018d, 10(2), 234.

Zhao Q.; Yao Y.; Yao W.; et al. Real-time precise point positioning-based zenith tropospheric delay for precipitation forecasting. *Scientific reports*, 2018b, 8(1), 7939-7939.

Zhao, Q., Yao, Y., Yao, W. Q., & Li, Z. Near-global GPS-derived PWV and its analysis in the El Niño event of 2014–2016. Journal of Atmospheric and Solar-Terrestrial Physics, 2018e, 179, 69-80.

---

## Referee Comment (RC2) · Anonymous Referee #2 · 9 Oct 2018

The paper investigates the water vapour variations, as derived from GNSS, during heavy rainfall events in Hong Kong. Both the 2-d Precipitable Water Vapour (PWV), and 4-d variations (from tomography) are studied. I find the study of the 4-d variations especially interesting.

I have already reviewed this paper when it was recently submitted to a different Journal. Since I see that the Authors did not modify the paper before submitting it to ANGEO, I will include also some major comments arisen in the second round of that review. I will refer all the comments to the document angel-2018-76-AC1-supplement.pdf.

[Figure]

General comments: the research design is not appropriate to make most of assessments; the results are not clearly presented and commented. The conclusions are not fully supported by the shown figures. The paper needs to include more comparison/validation data in order to evaluate the results.

The topic of the application of GPS tomography to study the water vapour variations during rainfall is definitively worth to be deeply investigated. However, such a study needs to include validation using an alternative technique or at least a detailed discussion on what variations one could expect. If that would be done, the study would also serve as a further validation of the GNSS tomography technique. In this paper it is hard to say if the detected variations are real or simply some artifacts of the tomographic solution. As a matter of fact, the comparison with radiosondes made in the paper (Fig 6) is not convincing. It is reasonable not expecting a perfect agreement between radiosondes and tomography for a number of reasons. But, the tomographic profiles show too many abrupt variations respect to the radiosonde profiles. It should be pointed out that the resolution of the radiosondes may not be as bad as it looks like. Commonly, a radiosonde samples with a relatively high vertical resolution (10-50 m). However, reported are normally only the values at heights where there are significant variations in at least one of the meteorological parameters, as well as for a number of predefined standard pressure levels. Hence, a big gap in the radiosonde data does not necessarily mean that data is missing, but rather that the variations of the meteorological parameters in this interval were rather smooth.

Under these circumstances the tomographic solution should be further checked in order to demonstrate that the temporal variations seen in the tomographic profiles are real (what is important for the conclusions of the paper).

Furthermore, I think the authors should also look more on the horizontal motions. For example, can it be shown that the vertical motions seen from the time variations of the vertical profiles are really vertical motions and not actually horizontal motions of sliding clouds?

Interactive
comment

[Figure]

The paper contains too many figures; I think same of them are not necessary.

Line 56-59 Use NWP only

Line 180 The "post-processing mode" means that you are using corrections for GPS data not available in real time. What is the data latency of final products downloaded from the ftp site? This could limit very much the use of this technique for nowcasting and/or assimilation in NWP. If you agree this limitation should be included and discussed in Conclusion section.

Line 239-240 "The weather conditions are cloudy and sunny without rainfall happened for the period of 1 to 8, August 2015" —> It would be to describe better the atmospheric conditions and the type of occurred events.

Line 240-241 " Those two periods are selected in this paper to investigate the variation characteristics of atmospheric water vapor" —> It can be useful to introduce in the article a graph related to the total reference period

Line 254 See comment to 180

Line 261-261 "PWV does not show any continuous increasing trend when there is no rainfall" —> Is it a common behaviour within the whole database or does it happen only in the period you are reporting about (1 to 8, August 2015)?

Fig. 2 and 4. The chosen time resolution does not allow making some conclusive assessments. The reader cannot evaluate from the figures if the precipitation effectively reduce the available PWV or how much is related with precipitation. What is the situation the days before the selected events that was generating high PWV values? Some meteorological description of the selected periods would help very much to understand.

Line 317 Since the main focus is the tomographic 4-d, the amount of introductory figures on 2-d could be too many.

Figure 7 to 10 —> Why so many panels? It would be better to reduce and compact the

figures, i.e. by including in one single panel the temporal sequence of profiles. Also the description should be improved.

---

## Author Comment (AC5) · 13 Oct 2018

The following is a point-to-point response to the reviewer's comments.

Response to Referee #2:

The paper investigates the water vapour variations, as derived from GNSS, during heavy rainfall events in Hong Kong. Both the 2-d Precipitable Water Vapour (PWV), and 4-d variations (from tomography) are studied. I find the study of the 4-d variations especially interesting.

I have already reviewed this paper when it was recently submitted to a different Journal. Since I see that the Authors did not modify the paper before submitting it to ANGEO, I will include also some major comments arisen in the second round of that review. I will refer all the comments to the document angel-2018-76-AC1-supplement.pdf.

✓ Thanks for the reviewer's comments, actually, we have tried our best to modify the comments and suggestions proposed by two reviewers before submitting it to ANGEO, and the corresponding responses to two reviewer's comments have been uploaded with the manuscript when we first submitted this manuscript to ANGEO.

✓ Additionally, we appreciate for the reviewer's second review of this manuscript. we have tried our best to revise the manuscript according to the reviewer's comments and suggestions.

**General comments**

The research design is not appropriate to make most of assessments; the results are not clearly presented and commented. The conclusions are not fully supported by the shown figures. The paper needs to include more comparison/validation data in order to evaluate the results.

✓ Thanks for the reviewer's comments, this manuscript has been improved by reducing some contents about 2-d PWV variations and some figures have been removed. Figures 2 and 3 have been replaced to make it observed clearly. The results and conclusions have been revised according to the reviewer's suggestion. Additionally, the explanation about the comparison/validation of tomographic result has been added in section 4.2.

The topic of the application of GPS tomography to study the water vapour variations during rainfall is definitely worth to be deeply investigated. However, such a study needs to include validation using an alternative technique or at least a detailed discussion on what variations one could expect. If that would be done, the study would also serve as a further validation of the GNSS tomography technique.

In this paper it is hard to say if the detected variations are real or simply some artifacts of the tomographic solution. As a matter of fact, the comparison with radiosondes made in the paper (Fig 6) is not convincing. It is reasonable not expecting a perfect agreement between radiosondes and tomography for a number of reasons. But, the tomographic profiles show too many abrupt variations respect to the radiosonde profiles. It should be pointed out that the resolution of the radiosondes may not be as bad as it looks like. Commonly, a radiosonde samples with a relatively high vertical resolution (10-50 m). However, reported are normally only the values at heights where there are significant

variations in at least one of the meteorological parameters, as well as for a number of predefined standard pressure levels. Hence, a big gap in the radiosonde data does not necessarily mean that data is missing, but rather that the variations of the meteorological parameters in this interval were rather smooth. Under these circumstances the tomographic solution should be further checked in order to demonstrate that the temporal variations seen in the tomographic profiles are real (what is important for the conclusions of the paper).

✓ Yes, we agree with the reviewer's opinion that the study needs validation. Actually, we have performed the validation and the tomographic results have been compared with that from radiosonde and ECMWF ERA-Interim data in our previous studies (Yao and Zhao, 2016, 2017; Zhao et al., 2017; Zhao et al., 2018). Due to this paper is mainly focus on the analysis of 4-d water vapour variations vertically during the heavy rainfall events and a series of quality evaluations of vertical profiles derived from GNSS tomography reconstruction have been carried out by our research group for the research area in the previous studies, therefore, a more detailed comparison information is not presented in this paper. The corresponding description and explanation have been added in P13L300-305.

✓ In our opinion, the detected variations derived from tomographic technique are relatively real for some reasons: (1) it can be observed from Figure 4 that the tomographic profiles are consistent with that from the radiosonde data at most heights; additionally, some studies have been carried out to obtain the atmospheric water vapor field in Hong Kong using the tomography technique by our research group and a good performance of tomographic result has been verified (Yao and Zhao, 2016, 2017; Zhao et al., 2017; Zhao et al., 2018); (2) Yes, it can be found that some abrupt variations are existed derived from tomographic result in Figure 4, but it is related with the weather conditions. According to the reported from Hong Kong Observatory, it is rained during the period of 19 to 27 July, 2015, and the following Figure (Figure 1) gives the hourly precipitation of SSP rain gauge during the period of 20 to 23 July, 2015. This station is selected because it is close to the radiosonde station (45004). It can be observed from Figure 1 that it was rained at UTC 00:00 of 20, 21, 22 and 23, therefore, the profile fluctuations are relatively large at those times as presented in Figure 4. On the contrary, there was no rain happened at UTC 12:00 of 20, 21, 22 and 23, and the profile fluctuations are relatively small. (3) it can be observed from Figure 5 that the profile fluctuations are not always large, and their fluctuations correspond to the occurrence of rainfall. Therefore, we think the tomography-derived water vapor profiles are relatively real in this paper, at least it can reflect the general variation of water vapor in vertical direction, especially during the rainfall period.

✓ As for the problem of radiosonde data, we have analyzed and removed the data of some layers because of the errors in pressure, water vapor pressure or temperature following some principles (Wang et al., 2008; Wang et al., 2016). Therefore, the radiosonde profiles are relatively coarse. Additionally, we have checked the radiosonde files carefully, which downloaded from ftp://ftp.ncdc.noaa.gov/pub/data/igra/ and found that the vertical resolution is non-uniformed with the value ranges from several hundred meters to more than one kilometer even the outliers are not removed.

[Figure]

Figure 1. Hourly precipitation of SPP rain gauge during the period of 20 to 23 July, 2015

References:

Yao Y.; Zhao Q. Maximally Using GPS Observation for Water Vapor Tomography. IEEE Transactions on Geoscience and Remote Sensing, 2016, 54(12), 7185-7196.

Yao Y.; Zhao Q. A novel, optimized approach of voxel division for water vapor tomography[J]. Meteorology and Atmospheric Physics, 2017, 129(1), 57-70.

Zhao Q.; Yao Y.; Yao W. A troposphere tomography method considering the weighting of input information[C]//Annales Geophysicae. Copernicus GmbH, 2017, 35(6), 1327-1340.

Zhao Q., Yao Y., Cao X., Zhou F. and Xia P. An Optimal Tropospheric Tomography Method Based on the Multi-GNSS Observations. Remote Sensing, 2018, 10(2), 234.

Wang, J., & Zhang, L. (2008). Systematic errors in global radiosonde precipitable water data from comparisons with ground-based gps measurements. J Clim, 21(10), 2218-2238.

Wang, X., Zhang, K., Wu, S., Fan, S., & Cheng, Y. (2016). Water vapor‑weighted mean temperature and its impact on the determination of precipitable water vapor and its linear trend. Journal of Geophysical Research Atmospheres, 121(2).

Furthermore, I think the authors should also look more on the horizontal motions. For example, can it be shown that the vertical motions seen from the time variations of the vertical profiles are really vertical motions and not actually horizontal motions of sliding clouds?)

✓ We appreciate for the reviewer's suggestion. Only the vertical water vapor variations are analyzed in this manuscript for some reasons: (1) according to the recordings derived from the Hong Kong Observatory, the convectional rain happened during the tested period, which is characterized by the vertical motions of atmospheric water vapor, therefore, this paper focus on the atmospheric water

vapor motion in vertical direction during the rainfall period; (2) the primary advantage of GNSS tomography technique is that the three-dimensional atmospheric water vapor distribution can be obtained from the regional GNSS networks. The vertical motion of water vapor variations can be reflected, which is unavailable from the 2-d PWV information. (3) the horizontal motion of atmospheric water vapor has been widely investigated by previous studies while the vertical motion is rarely discussed before; (4) analysis of the horizontal motion of atmospheric water vapor is not the main point of this paper, therefore, we didn't investigate the horizontal motions.

✓ The vertical variations of water vapor profiles with time at different heights correspond to the occurrence of rainfall according to the analysis in Section 4.2, therefore, we think the time variations of the vertical profiles are really vertical motions.

Reference:
Yu, C., Penna, N. T., & Li, Z. (2017). Generation of real-time mode high-resolution water vapor fields from GPS observations. Journal of Geophysical Research: Atmospheres, 122(3), 2008-2025.

The paper contains too many figures; I think same of them are not necessary.
✓ Thanks for the reviewer's suggestion, some figures have been removed or combined and only 9 figures are left in the revised manuscript.

Line 56-59 Use NWP only.
✓ Thanks for the reviewer's reminding, this expression has been revised,

Line 180 The "post-processing mode" means that you are using corrections for GPS data not available in real time. What is the data latency of final products downloaded from the ftp site? This could limit very much the use of this technique for nowcasting and/or assimilation in NWP. If you agree this limitation should be included and discussed in Conclusion section.
✓ Yes, we haven't used the real-time products in this experiment. This is because: (1) the final products can be available from the ftp site, which is very convenient; (2) the date of experimental data is at the year of 2015; therefore, it is not necessary to use the real-time products. The data latency of final products is about two weeks.
✓ Currently, the real-time products of satellite orbit and clock corrections can also be obtained without time latency by our research group, and this experiment can also be performed using the real-time products. Therefore, we think this will not limit the using of tomographic technique for nowcasting and/or assimilation in NWP.

Line 239-240 "The weather conditions are cloudy and sunny without rainfall happened for the period of 1 to 8, August 2015" —> It would be to describe better the atmospheric conditions and the type of occurred events.

✓ We appreciate for the reviewer's reminding, and we are sorry that we cannot describe more about the atmospheric conditions for the period of 1 to 8, August 2015 currently. Because no more atmospheric conditions information can be acquired currently according to the recording of Hong Kong Observatory (HKO), and we have tried to contact somebody in HKO to obtain more information about the weather information during this period.

Line 240-241 " Those two periods are selected in this paper to investigate the variation characteristics of atmospheric water vapor" —> It can be useful to introduce in the article a graph related to the total reference period.
✓ Thanks for the reviewer's reminding, this sentence has been moved to the beginning of Section 3.1 (P6L167-168).

Line 261-261 "PWV does not show any continuous increasing trend when there is no rainfall" —> Is it a common behaviour within the whole database or does it happen only in the period you are reporting about (1 to 8, August 2015)?
✓ We appreciate for the reviewer's question, in our opinion, this is a common behavior when there is no rain happened.
✓ Actually, we have done some experiments and found that there is a continuous increasing trend when there is the rainfall happened, but this phenomenon is not happened when there is no rainfall occurred, for example in the follow Figure, it can be clearly seen that a continuous increasing trend before the occurrence of rainfall is appeared but not obvious when there is no rain happened.

[Figure]

Figure. Hourly precipitation and PWV time series for four stations (ZHOS, ZJXC, LJSL and ZJPH) over different periods

Fig. 2 and 4. The chosen time resolution does not allow making some conclusive assessments. The reader cannot evaluate from the figures if the precipitation effectively reduces the available PWV or how much is related with precipitation. What is the situation the days before the selected events that was generating high PWV values?

Some meteorological description of the selected periods would help very much to understand.

✓ Thanks for the reviewer's suggestion, the Figure 2 has been revised and only the time period of 19 to 21 July, 2015 are presented to better describe the relationship between PWV and rainfall, and the corresponding content has been added in P10L246-249. Additionally, weather condition has been added in the first paragraph of Section 4.

Line 317 Since the main focus is the tomographic 4-d, the amount of introductory figures on 2-d could be too many.

✓ Thanks for the reviewer's suggestion, some figures and description about the 2-d PWV variations have been removed in the section 4.1. In the revised manuscript, only two figures left in the section of 2-d PWV analysis.

Figure 7 to 10 —> Why so many panels? It would be better to reduce and compact the figures, i.e. by including in one single panel the temporal sequence of profiles. Also the description should be improved.

✓ Thanks for the reviewer's question, two successive water vapor profiles are presented in a single panel in order to compare the continuous variations of water vapor profiles with time conveniently. Therefore, there are some panels in a figure. By this way, the continuous water vapor variation with time can be clearly observed during the experimental period. In addition, the corresponding description about Section 4.2 has been improved according to the reviewer's suggestion.

We appreciate for Reviewer's warm work earnestly, which has a significant improvement for our manuscript. And we hope that our corrections meet with the reviewer's requirement. Once again, thank you very much for your comments and suggestions.

---

## Author Comment (AC6) · 13 Oct 2018

The comment was uploaded in the form of a supplement:
https://www.ann-geophys-discuss.net/angeo-2018-76/angeo-2018-76-AC6-supplement.pdf